# LogiCoT: Logical Chain-of-Thought Instruction Tuning

**Hanmeng Liu**
Zhejiang University
liuhanmeng@zju.edu.cn

**Zhiyang Teng**[*]
Nanyang Technological University
zhiyang.teng@ntu.edu.sg

**Leyang Cui**
Tencent AI lab
leyangcui@tencent.com

**Chaoli Zhang**
Alibaba Group
chaoli.zcl@alibaba-inc.com

**Qiji Zhou** and **Yue Zhang**[*]
Westlake University
{zhouqiji, zhangyue}@westlake.edu.cn

## Abstract

Generative Pre-trained Transformer 4 (GPT-4) demonstrates impressive chain-of-thought reasoning ability. Recent work on self-instruction tuning, such as Alpaca, has focused on enhancing the general proficiency of models. These instructions enable the model to achieve performance comparable to GPT-3.5 on general tasks like open-domain text generation and paraphrasing. However, they fall short of helping the model handle complex reasoning tasks. To bridge the gap, this paper presents LogiCoT, a new instruction-tuning dataset for **Logi**cal **C**hain-**o**f-**T**hought reasoning with GPT-4. We elaborate on the process of harvesting instructions for prompting GPT-4 to generate chain-of-thought rationales. LogiCoT serves as an instruction set for teaching models of logical reasoning and elicits general reasoning skills.

## 1 Introduction

Instruction tuning Large Language Models (LLMs) has become a popular paradigm for Natural Language Processing (NLP) in recent years (Ouyang et al., 2022; Sun et al., 2022). A prominent line of research is the development of OpenAI's ChatGPT and GPT-4 (OpenAI, 2023). LLMs demonstrate multi-step chain-of-thought (CoT) reasoning ability with proper prompting (Kojima et al., 2022; Huang and Chang, 2022). CoT instruction tuning has drawn attention for its potential to encourage complex, step-by-step reasoning. For example, Wei et al. (2023) and Kojima et al. (2022) have demonstrated the ability of LLMs to generate a coherent sequence of reasoning steps leading to the final answer through CoT prompting. Moreover, ChatGPT and GPT-4 have shown remarkable zero-shot complex reasoning abilities on several logical reasoning datasets (Liu et al., 2023a).

Yet, developing such proprietary models as GPT-4 and ChatGPT often necessitates intensive data

---

[*]Corresponding Authors

Figure 1: A showcase of using GPT-4 and existing inference data to generate CoT rationales for logical reasoning.

and instruction engineering, a process that has, thus far, been largely kept private. Recent research endeavours have begun to explore the distillation of the instruction data using self-instruct techniques (Wang et al., 2022; Peng et al., 2023), where GPT-3 or GPT-4 are used to generate instruction-following examples. This technique represents a promising avenue for reducing the human labour involved in instruction tuning, offering a more economical way to produce community models trained with instructional data. A paradigmatic example is the pipeline by Wang et al. (2022) for cultivating instruction data, where initial instructions are authored by humans, and LLMs are then used to extend this instruction data. This pipeline has been used to produce multiple open-sourced, instruction-tuned

models, such as Alpaca (Taori et al., 2023) and Vicuna (Chiang et al., 2023). The instruction tuning data share a common pool of instructions and fixed task templates. However, the scope of these instructions is limited and does not encapsulate diverse, complex reasoning scenarios such as multi-step logical reasoning.

Logical reasoning is crucial in human cognition, embodying the ability to infer conclusions based on a structured progression of premises. The dearth of such abilities in community models (Liu et al., 2023a) presents a significant gap, inhibiting the development of open LLMs with strong chain reasoning. While GPT-4 has demonstrated its ability to produce high-quality CoT reasoning output, the potential of generating CoT instruction tuning data using this model remains largely unexplored. The need to cover more diverse and complex reasoning scenarios, particularly multi-step logical reasoning, represents a significant gap in the current instruction-tuning landscape.

We aim to address this gap by scaling up the instruction set (Chung et al., 2022), paving the way for more nuanced and sophisticated instruction-tuned models. To this end, we introduce LogiCoT, a chain-of-thought (CoT) instruction-tuning dataset designed explicitly for logical reasoning. Our approach involves repurposing existing logical reasoning datasets, constructing logical reasoning CoT instructions from these resources, and leveraging the capabilities of GPT-4 to generate high-quality outputs. The resulting instruction data features both symbolic reasoning and multi-step CoT reasoning, providing a comprehensive and nuanced resource for enhancing the logical reasoning abilities of AI models.

Using LogiCoT, we fine-tune a LLaMA-7b (Touvron et al., 2023) model with an instruction-tuning scheme, resulting in a small-sized instruction-tuned open model. Results on both logical reasoning benchmarks and general human-centric benchmarks indicate remarkable performance elevations compared to state-of-the-art instruction-tuned models, showing the promise of leveraging complex CoT instruction data in the instruction-tuning process of LLMs.

Our work is in line with recent research indicating that smaller language models can achieve competitive multi-step reasoning abilities when specialized on targeted CoT tasks. Examples of these tasks include the execution of SQL commands, mathe-

matical CoT reasoning, and generating code snippets (Gao et al., 2022; Liu et al., 2023b; Fu et al., 2023). By applying a similar specialization approach to the broader and more complex domain of logical reasoning, we aim to bring these capabilities into the mainstream, furthering the development of AI systems with advanced logical reasoning skills.

We release both the training data and model weights.

## 2 Related Work

**Instruction tuning LLMs.** Instruction-tuning of Large Language Models (LLMs) has become a thriving research area in Natural Language Processing (NLP), aiming to enable zero-shot generalization on unseen tasks (Zhong et al., 2021; Ouyang et al., 2022; Wei et al., 2022). This involves fine-tuning LLMs to perform diverse tasks by following a set of instructions, making the task source an essential component of instruction tuning (Longpre et al., 2023). Most existing instruction tuning methods rely heavily on human-crowdsourced tasks or model-generated tasks. Human-crowdsourced tasks originate from a variety of sources, such as T0 (Sanh et al., 2022), FLAN (Wei et al., 2022), and NaturalInstructions (Mishra et al., 2022). These tasks, although high-quality, rely on substantial human effort and are often limited in quantity. In contrast, model-generated tasks involve leveraging a powerful language model, such as GPT-3 and GPT-4, to generate a diverse set of instructions, task inputs, and task outputs based on a seed set (Wang et al., 2022; Peng et al., 2023). We seek to leverage GPT-4's chain-of-thought reasoning capabilities in instruction tuning. By introducing the LogiCoT dataset and incorporating symbolic tasks, we aim to advance the quality and scalability of instruction-following data and thereby enhance the overall performance of instruction-tuned LLMs.

PINTO (Wang et al., 2023) offers a two-fold approach to elucidate the reasoning process of large language models (LMs). The core innovation lies in its prompt-based learning combined with counterfactual regularization to ensure faithfulness in the reasoning over generated rationales. While PINTO excels in achieving superior performance across in-distribution and out-of-distribution test sets and ensuring that rationales are faithful, its primary motivation is different from LogiCoT. Whereas PINTO emphasizes rationale transparency and faithfulness, LogiCoT is geared towards using

logic-related chain-of-thought data to achieve advanced logical reasoning skills.

Teaching small language models to reason is another research direction (Magister et al., 2023; Li et al., 2023; Hsieh et al., 2023) that underscores the significance of knowledge distillation in porting reasoning capabilities from mammoth models to their relatively compact counterparts. Central to this approach is the fine-tuning of smaller models on the chain-of-thought rationales spawned by larger models. Such a strategy has demonstrated marked improvements in various reasoning datasets. Yet, this is where LogiCoT diverges. Our approach uniquely employs logical reasoning rationales and capitalizes on the generative prowess of GPT-4.

In essence, while the broader arena is populated with innovative methodologies aiming to amplify the reasoning prowess of LMs, LogiCoT distinguishes itself by synergizing logical reasoning with the generative might of GPT-4, setting a precedent in logical reasoning tasks.

**Chain-of-thought rationales.** Large Language Models (LLMs) can conduct complex reasoning tasks by generating intermediate reasoning steps through a process called chain-of-thought (CoT) prompting (Wei et al., 2023). Zero-Shot-CoT prompting uses a simple instruction (like "Let's think step by step") to elicit step-by-step reasoning before answering a question. LLMs have exhibited reasonable zero-shot reasoning capabilities, generating outputs that inherently reflect CoT reasoning (Zhou et al., 2023). This notion inspired researchers to use self-generated rationales for demonstrations. In particular, Zelikman et al. (2022) demonstrated the practicality of using LLMs to generate rationales. They prompted GPT-J (Wang and Komatsuzaki, 2021) to generate rationales and then selected the ones leading to the correct answer. We adopt this method for data collection using GPT-4. Our approach, however, focuses on complex logical reasoning scenarios using questions with annotated answers.

**Logical reasoning.** Logical reasoning is a key aspect of human cognition and a critical capability for AI systems. Researchers have been exploring various approaches to achieve this goal, including rule-based methods and symbolic systems (MacCartney and Manning, 2007), fine-tuning large language models (Wang et al., 2018), and combining both neural and symbolic approaches (Li and Sriku-

mar, 2019). Logical reasoning tasks can require multi-step, complex reasoning, which makes them an ideal target for CoT instruction tuning. To our knowledge, we are the first to consider this method for logical reasoning, making use of rich reasoning chain data finetuning LLMs, increasing their task performance.

# 3 Dataset

The data construction of LogiCoT is a multistage process that uses GPT-4 as a teaching assistant, which is illustrated in Figure 1. We first establish a foundation by choosing suitable seeding data with gold output and optional CoT reasoning chains. With seeding data in place, we proceed to formulate instructions. The process involves translating the intended tasks into clear, unambiguous prompts that elicit GPT-4's capacity for logical reasoning rationales. We then combine the seeding data and the corresponding instructions and feed them into GPT-4 to generate responses. GPT-4's output is guided by the gold label and the reasoning chain. We use both the gold output and GPT-4 response as our instruction data. Figure 1 illustrates this process.

## 3.1 Seminal Data Selection

Selecting the seminal data for CoT instruction tuning of logical reasoning models involves choosing high-quality datasets that adequately cover the range of skills required for logical reasoning. The datasets should present challenges representing real-world logical reasoning tasks and be designed to support CoT instruction tuning. Below are the seminal instruction data we choose:

**LOGICINFERENCE** (Ontanon et al., 2022) is a synthetically generated sequence-to-sequence dataset teaching models to perform logical inference using propositional logic and a subset of first-order logic. Figure 2 in Appendix A shows an example. The input is a question, varying in types of problems ranging from language-to-logic translation to multi-step inference chains. The output provides the answer, including the reasoning chain to generate it. The output, in some cases, even provides the name of the inference rule used in each step.

**EntailmentBank** (Dalvi et al., 2021) is an open-domain question answering data with rationales as an entailment tree. It uses multiple-choice questions from grade school science. The entailment

trees are constructed with human labour. Figure 3 in Appendix A shows an example. EntailmentBank provides a unique combination of open-domain question answering and logical reasoning in a format that closely aligns with our instruction tuning objectives. The structure of its rationales, presented as entailment trees, gives our model the opportunity to learn from and adapt to complex logical relationships in a controlled environment.

**FOLIO** (Han et al., 2022) is an open-domain, logically complex and diverse dataset equipped with first-order logic (FOL) annotations. Figure 4 in Appendix A shows an example. What sets FOLIO apart is the parallel FOL annotations for each premise and conclusion, which are automatically verified by an FOL inference engine. This aspect provides a clear, precise standard for logical reasoning. In addition, the human-annotated nature of the dataset ensures high-quality data input. This dataset can be easily converted into a sequence-to-sequence structure, serving as instruction-following data for symbolic logic reasoning.

**ReClor** (Yu et al., 2020) and **LogiQA** (Liu et al., 2020) are datasets derived from verbal reasoning examinations, demanding various types of logical reasoning for answering multi-choice questions. Figure 5 in Appendix A shows an example from the ReClor data. These datasets are especially valuable in that they represent realistic human reasoning processes. Further, the real-world nature of the questions in these tests, which often require a mix of common sense and logical reasoning, ensures that the model is trained to tackle problems with varying degrees of complexity. We use the training set of the two datasets, keeping the test set out of the instruction tuning data. Specifically, we use the Chinese version of the LogiQA dataset.

These seminal datasets for CoT instruction tuning in GPT-4 offer a balanced, comprehensive, and challenging training environment. This approach ensures that the model gains exposure to a broad range of logical reasoning tasks, thus enhancing its ability to effectively handle similar tasks in real-world applications.

## 3.2 Instruction Types

We consider different types of instructions for instructing language models in various aspects of logical reasoning. Each type is designed to engage the model with logical inference tasks at differ-

ent levels of abstraction and complexity, with both natural language and symbolic language. To our knowledge, no similar instruction types exist in other instruction-following data.

We classify the instruction types into general inference (Section 3.2.1) and multi-choice reading comprehension (Section 3.2.2) tasks.

### 3.2.1 General Inference Task

This category includes instruction types that demand general reasoning and inferential skills, often involving an understanding of logical structures and principles. The model may need to perform operations such as translating natural language to formal logic, predicting possible inferences from given premises, or tracing inference chains. These tasks are designed to enhance the model's ability to think critically and logically, without relying too heavily on specific contexts or domain knowledge.

Table 7 in Appendix B shows the instruction types for general inference. An example is offered to illustrate each instruction type.

**Language to Logic**: This instruction involves translation from natural language into a more formal logical notation. It presents a foundational task of understanding and interpreting logical statements expressed in natural language and converting them into a formalized logical representation.

**One-Step Inference**: In this case, the model is presented with a set of premises and tasked with predicting all the potential inferences that can be derived from them in a single step. This type of instruction encourages the model to exercise deductive reasoning based on the provided premises. The premises and inferences can be in natural language or symbolic language. The latter encourages precise and abstract reasoning, while the former context simulates real-world language use scenarios.

**Inference Chains**: This instruction type takes logical reasoning a step further by requiring the model to establish whether a potential inference can be proven from a set of premises. The model must then provide the chain of reasoning leading to the answer. This type encourages deeper logical reasoning and the ability to construct logical arguments. The examples are either crafted in symbolic language or natural language.

### 3.2.2 Reading Comprehension Tasks

Machine reading comprehension (MRC) is the go-to task for testing LLMs' reasoning ability, where a

model is given a passage and a question and asked to find the answer. This category involves tasks that require a deep understanding of a given text, often demanding that the model identifies, extracts, or infers information from the text. The model might be asked to resolve a situation described in the text, to pinpoint a flaw in an argument presented, or to identify information that would either strengthen or weaken an argument.

Table 8 in Appendix B shows the instruction types and running examples for logical reading comprehension.

The distinctiveness of these instruction types lies in their combination of logical reasoning tasks with natural language processing, providing a robust framework for training language models in logic-infused language understanding and generation. This comprehensive approach is unique to this data generation scheme and offers an innovative pathway for improving the logical reasoning capabilities of large language models.

## 3.3 Data Collection

We are granted early access to GPT-4 API, which provides a unique opportunity to leverage the advanced capabilities of this model for generating high-quality rationales. By using the API, we can pass the logical reasoning tasks derived from our seminal datasets (LogiQA, ReClor, LOGICIN-FERENCE, and FOLIO) to GPT-4 and collect the model's responses. The pseudo-code in Appendix C exemplifies the process for GPT-4 rationales collection.

The general inference task is converted from LOGICINFERENCE, EntailmentBank, and FOLIO. The three datasets are particularly valuable in this context as they offer data instances accompanied by precise reasoning chains. These reasoning processes, whether derived from rules or written by humans, serve as concrete examples for GPT-4 to learn from. Golden CoT output datasets provide GPT-4 with invaluable references, optimizing generation quality.

The machine reading comprehension task is derived from LogiQA and ReClor. These two datasets are not sequence-to-sequence; they do not offer step-by-step reasoning processes. However, GPT-4 scores well on these two datasets (Liu et al., 2023a) without in-context examples, GPT-4's commendable performance on these datasets assures quality generation.

| Task | Origin | Size |
|---|---|---|
| Language to Logic | LOGICINFERENCE & FOLIO | 13,206 |
| One-Step Inference | LOGICINFERENCE & FOLIO | 23,943 |
| Inference Chain | LOGICINFERENCE & EntailmentBank | 26,228 |
| Multi-choice | LogiQA & ReClor | 5,606 |

Table 1: Data statistics.

## 3.4 Data Statistics and Quality Assessment

We collected 68,983 data instances in total. We illustrate their distribution in Table 1. Table 9 in Appendix D shows the root verb-noun pairs of our instruction set. As can be seen from the figure, the instructions are reasoning-centered. Figure 6 in Appendix D shows the root verb-noun pairs of GPT-4 responses, which cover a wide range of scenarios both in everyday language and in symbolic language.

Together, these tasks cover various logical reasoning abilities. We release our LogiCoT instruction tuning dataset at https://huggingface.co/datasets/csitfun/LogiCoT to facilitate future research.

**Human Evaluation of Dataset Quality** We conducted a comprehensive human evaluation of our generated reasoning chains. A random subset of 200 reasoning chains was selected. These chains were evaluated by 3 domain professional annotators using four key metrics: Relevance (Does the chain directly relate to the question? ), Coherence (Is the chain logically consistent? ), Completeness (Does it offer a full explanation for the reasoning?) , and Faithfulness ( Is the reasoning factual and not fabricating details? ). Each reasoning chain was rated on a scale from 1 (poor) to 5 (excellent) for each metric.

The reasoning chains achieved an average score of 4.9 for Relevance, 4.7 for Coherence, 4.5 for Completeness, and 4.5 for Faithfulness. The inter-annotator agreement, measured using Cohen's Kappa, was 0.87, indicating strong agreement among the annotators.

The human evaluation underscores the quality and reliability of the reasoning chains generated by our model. While the scores were high for relevance, we acknowledge room for improvement in faithfulness, and future work will aim to refine the generation process to address this.

## 4 Experiments

We use LogiCoT to fine-tune an open LLM and test the resulting model on logical reasoning and general human-centric benchmarks.

| Dataset | LQ | LQ zh | RC | AL | LQ ood | CT | HL | TN | Overall |
|---|---|---|---|---|---|---|---|---|---|
| **Size** | 1572 | 1594 | 500 | 230 | 1354 | 805 | 35891 | 10071 | 52017 |
| LLaMA-7b-base | 18.04 | 19.06 | 15.83 | 13.91 | 20.25 | 32.40 | 25.20 | 37.35 | 20.22 |
| LLaMA-30b-supercot | 19.31 | 26.35 | 17.81 | **17.98** | 18.41 | 24.10 | 32.26 | 41.91 | 24.78 |
| Falcon-40b-instruct | 23.21 | 19.77 | 26.77 | 12.70 | 17.33 | 16.13 | 28.49 | 44.66 | 23.63 |
| LLaMA-7b-logicot | **50.25** | **32.77** | **57.60** | 16.96 | **38.79** | **35.68** | 35.44 | **58.05** | **40.69** |

Table 2: LogiEval results. *LQ*: LogiQA 2.0, *RC*: ReClor, *AL*: AR-LSAT, *CT*: ConTRoL, *HL*: HELP, *TN*: TaxiNLI. All results are in %, the best ones are in **bold**, and the second best ones are in underline.

## 4.1 Models

**LLaMA** (Touvron et al., 2023) is an open-sourced LLM developed by Meta. We adopt LLaMA-7b (Touvron et al., 2023) as the base model for instruction tuning.

We shuffle the collected data to ensure the base model encounters a broad range of data characteristics throughout the instruction tuning phase. This approach bolsters the model's potential to generalize its learning to a variety of unseen situations. Following (Taori et al., 2023), we adopted their running scripts and set the learning rate to 2e-5, and the batch size to 4. We trained 2 epochs. Our experimental setup incorporated two A100 GPUs, running the instruction tuning process over 2 epochs. We use the Microsoft deepspeed library [1] to accelerate the training process. The training takes 4 days. We release our instruction-tuned LLaMA model as "LLaMA-7b-logicot".

To compare our model performance with other open-sourced instruction-tuned LLMs, we choose two top models from the Open LLM Leaderboard [2], namely LLaMA-30b-supercot and Falcon-40b-instruct. The former is a LLaMA-30b model tuned on the SuperCOT dataset, [3] which is also a CoT dataset. The latter is an instruction-tuned **Falcon-40b** model (Almazrouei et al., 2023) tuned on a mix of datasets. [4]

## 4.2 Benchmarks

**Logical reasoning** The purpose of this exercise is to directly measure how effectively our model has incorporated the logical reasoning skills learned during the instruction tuning phase. This evaluation helps us understand the strength of our model in the domain it was explicitly tuned for.

| Dataset | Target |
|---|---|
| LogiQA 2.0 test | 4-way multi-choice |
| LogiQA 2.0 zh test | 4-way multi-choice |
| ReClor dev | 4-way multi-choice |
| AR-LSAT test | 5-way multi-choice |
| LogiQA 2.0 OOD | 4-way multi-choice |
| ConTRoL test | E, C, N |
| HELP test | E, C, N |
| TaxiNLI test | E, C, N |

Table 3: Data statistics. ("E" refers to "entailment"; "C" refers to "contradiction"; "N" refers to "neutral".)

For the assessment of our model's logical reasoning capabilities, we select the LogiEval benchmark (Liu et al., 2023a) as our primary testing suite, which is expressly designed with an instruct-prompting style to rigorously test the logical reasoning abilities of large language models (LLMs). Each dataset instance has integrated instructions which ensures the LLM comprehends the desired format and response requirements. Table 3 illustrates the 8 datasets and their task formats in the LogiEval benchmark.

The LogiEval benchmark consists of two types of tasks. Firstly, it encompasses multi-choice reading comprehension tasks that examine the model's ability to interpret, analyze, and make decisions based on given texts. Secondly, it includes natural language inference tasks, presenting an opportunity for the model to demonstrate its understanding of logic and coherence in language constructs. This combination of tasks within LogiEval provides extensive coverage of logical reasoning tasks, making it an ideal choice for our testing purposes.

**General Tasks** We further go beyond the confines of logical reasoning tasks. We probe the model's capabilities on general human-centric language model benchmarks. This broad-based testing strategy is aimed at gauging the generalizability of our model. It enables us to measure the performance of our model on tasks that span a wider array of human language processing tasks, beyond its specific training focus.

To conduct a comprehensive evaluation of our model, we employ the Massive Multitask Language

---

[1] https://github.com/microsoft/DeepSpeed
[2] https://huggingface.co/spaces/HuggingFaceH4/open_llm_leaderboard
[3] https://huggingface.co/datasets/kaiokendev/SuperCOT-dataset
[4] https://huggingface.co/tiiuae/falcon-40b-instruct

| Task | LLaMA-7b-base | LLaMA-7b-logicot |
|---|---|---|
| Math | 25.1 | **29.0** |
| Health | 34.0 | **42.9** |
| Physics | 29.4 | **34.2** |
| Business | 34.8 | **57.2** |
| Biology | 32.4 | **46.5** |
| Chemistry | 25.4 | **33.0** |
| Computer science | 28.2 | **40.0** |
| Economics | 26.5 | **38.5** |
| Engineering | 25.5 | **32.4** |
| Philosophy | 30.8 | **37.6** |
| Other | 40.4 | **50.8** |
| History | 38.2 | **55.2** |
| Geography | 28.8 | **52.5** |
| Politics | 32.1 | **53.4** |
| Psychology | 33.2 | **50.9** |
| Culture | 37.3 | **56.9** |
| Law | 29.3 | **39.6** |
| STEM | 27.6 | **34.8** |
| Humanities | 31.6 | **41.8** |
| Social Sciences | 31.5 | **49.2** |
| Other (misc.) | 36.4 | **47.7** |
| Average | 31.8 | **43.3** |

Table 4: MMLU results for LLaMA-7b base and logicot. All results are in %, the best results are in **bold**.

Understanding (MMLU) benchmark (Hendrycks et al., 2021). This benchmark assesses a large language model's capabilities across a diverse range of domains, spanning from foundational knowledge sectors such as mathematics and history, to more specialized fields like law and ethics. We remain consistent with the protocols from previous evaluations.

Our evaluation approach emphasizes exact matching. We instruct LLMs to generate the precise answer in accordance with the given prompt. The first token of the output is deemed as the final answer, which tests the LLM's proficiency in following instructions. The instruction prompts for the two tasks are detailed in Appendix E.

## 4.3 Results

The results on the LogiEval benchmark are shown in Table 2. With the varies of different data sizes, we represent the data size of each incorporated dataset in the second row. We report the performance of LLaMA-7b base model as one of the baselines.

Compared to the LLaMA-7b base model, LLaMA-7b-logicot surpasses the base model on all datasets. Specifically, LLaMA-7b-logicot performs much better on the multi-choice reading comprehension task, except for the AR-LSAT data, where all four models give a relatively low performance. Compared to the two leading open-sourced models, our model gives the best performance except for the AR-LSAT data, where our model yields 16.96%

accuracy, the second best compared to 17.98% accuracy of the LLaMA-30b-supercot model. Notice that the AR-LSAT dataset is a 5-way classification task, in contrast to our instruction tuning data's 4-way multi-choice reading comprehension.

The results on the MMLU benchmark are shown in Table 4. Overall, the average accuracy of LLaMA-7b-logicot is 43.3%, compared to the 31.8% accuracy of the LLaMA-7b base model. The improvement is salient given that our model has not been specifically tuned on the tested subjects. In detail, our model surpasses LLaMA-7b-base on every subject. The best improvements are seen in business, computer science, economics, etc., with over 10 points accuracy boost.

## 4.4 Discussion

### 4.4.1 Compare to ChatGPT and GPT-4

As our model is tuned on instruction data distilled from GPT-4, we therefore compare their performance on the LogiEval benchmark with our model. The results are shown in Tabel 5. LLaMA-7b-logicot performs on par with ChatGPT on datasets like LogiQA 2.0 and AR-LSAT. Particularly, on ReClor, LogiQA OOD, and TaxiNLI, our model outperforms ChatGPT by a small margin. Given that LLaMA-7b-logicot is a small model with only 7 billion parameters, the results are quite inspiring. However, on the Chinese version of LogiQA, and on ConTRoL, the performance of our model lags ChatGPT by 20 points. This performance discrepancy suggests that our model may exhibit weaknesses in handling Chinese corpus and passage-level texts.

GPT-4 outperforms LLaMA-7b-logicot on each dataset, giving the best performance on the LogiEval benchmark except for the ConTRoL dataset, where the accuracy is lower than ChatGPT by 2 points. The results show that there is still a gap between our model and the best proprietary model.

### 4.4.2 Ablation Study

To better understand the specific contributions of various reasoning types employed in our instruction-tuning process, we conduct an ablation study. This involves evaluating model performance by selectively removing one reasoning type at a time and observing the resultant change in the model's ability to handle logical reasoning tasks.

The ablation study uses the instruction-tuned LLaMA-7b model. For each reasoning type, we

| Dataset | LQ | LQ zh | RC | AL | LQ ood | CT | HL | TN | Overall |
|---|---|---|---|---|---|---|---|---|---|
| LLaMA-7b-logicot | 50.25 | 32.77 | 57.60 | 16.96 | 38.79 | 35.68 | 35.44 | 58.05 | 40.69 |
| ChatGPT | 52.37 | 53.18 | 57.38 | 20.42 | 38.44 | **58.45** | 42.13 | 57.30 | 47.46 |
| GPT-4 | **72.25** | **70.56** | **87.20** | **33.48** | **58.49** | 56.40 | **46.01** | **60.08** | **60.56** |

Table 5: ChatGPT and GPT-4 on LogiEval. *LQ*: LogiQA 2.0, *RC*: ReClor, *AL*: AR-LSAT, *CT*: ConTRoL, *HL*: HELP, *TN*: TaxiNLI. All results are in %, the best ones are in **bold**, and the second best ones are in underline.

| Removed | LogiEval | MMLU |
|---|---|---|
| None (Full data) | 40.7 | 43.3 |
| Language to Logic | 32.4 | 38.5 |
| One-step Inference | 38.1 | 37.7 |
| Inference Chain | 30.8 | 35.0 |
| Multi-choice | 35.6 | 30.9 |

Table 6: Ablation study by removing reasoning types. Results in accuracy (%).

train the model without that specific type while retaining the others. This is performed iteratively for all reasoning types.

We report the overall average score of LogiEval and MMLU, respectively, to provide a comprehensive understanding of the respective impacts of these reasoning types. The results are shown in Table 6. We elaborate on the results of ablating "multi-choice" in Table 10 of Appendix F to give audiences a clear view.

**Language to Logic:** Excluding this type led to a decline in performance on the LogiEval dataset. This underscores its significance in mapping linguistic constructs to logical expressions, which is foundational for reasoning tasks.

**One-step Inference:** The drop in accuracy suggests that even simple inferences play a vital role, especially in tasks where direct conclusions are drawn from given premises.

**Inference Chain:** The model's performance drop on both datasets highlights the importance of chained logical deductions. Tasks that require multi-step reasoning particularly benefit from this type.

**Multi-choice:** Removing multi-choice reasoning impacted performance on MMLU more than on LogiEval, emphasizing its role in tasks where choosing among alternatives based on logical grounds is essential.

This ablation study reaffirms the unique contributions of each reasoning type to the model's performance. While all reasoning types contribute to enhancing the model's logical understanding, their impacts vary based on the nature of tasks and the datasets used. Future research can delve deeper into optimizing instruction-tuning processes based on specific reasoning type requirements of datasets.

### 4.4.3 Case Study

To further understand the reasoning ability of LLaMA-7b-logicot, we provide case study examples in this section. Rather than generating a single output token, we ask the model to generate both the answer and rationales by prompting it with "Answer and reason: ".

Figure 7 in Appendix G shows an example of LLaMA-7b-logicot solving the MRC task correctly. In this case, our model is asked to find the option that strengthens the given argument. The key idea in this example is that to save costs, the company needs to abandon a rule that leads to higher costs. Our model successfully identifies option A as proof of the costly rule. It also generates a coherent and convincing explanation.

Figure 8 in Appendix G shows an example of LLaMA-7b-logicot solving the MRC task incorrectly. The example is a difficult analytical question involving complex logical relations and numerical calculations. The reasoning steps generated by our model are sensible in (1), (2), and (3). However, it came up with new constraints, as in (4), (5), (6), and (11). Moreover, the model does not examine all the options to decide the correct answer. Our model made a series of mistakes in this example which led to the wrong conclusion.

## 5 Conclusion

We collected LogiCoT, a set of CoT instruction-tuning data using GPT-4 through the lens of logical reasoning tasks. With 70K training instances, LogiCoT offers diverse challenges in logical reasoning. Using LogiCoT, we instruction-tuned an open-sourced LLaMA-7b model and the experiments on our model demonstrate competitive logical reasoning and general inference abilities. To our knowledge, we are the first to construct an instruction-tuning dataset with rich and diverse logical reasoning steps, showing its potential to enhance a generative LLM. Future work includes the integration of our dataset to enhance dialogue-oriented LLMs such as Alpaca.

## Acknowledgement

We thank all the anonymous reviewers for their constructive suggestions. This publication has emanated from research conducted with the financial support of the Pioneer and "Leading Goose" R&D Program of Zhejiang under Grant Number 2022SDXHDX0003. Zhiyang Teng is partially supported by CAAI-Huawei MindSpore Open Fund (CAAIXSJLJJ-2021-046A). Yue Zhang and Zhiyang Teng are the corresponding authors.

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

Input:
What can be inferred from the following premises in a single inference step (ignoring inferences that add new predicates or constants)?
David is working if and only if it is cloudy.
David is not working.

Output:
If David works, then it is cloudy.
If it is cloudy, then David is working.
It is not cloudy.

Figure 2: An instance from the LOGICINFERENCE dataset.

*2018 EMNLP Workshop BlackboxNLP: Analyzing and Interpreting Neural Networks for NLP.*

Ben Wang and Aran Komatsuzaki. 2021. GPT-J-6B: A 6 Billion Parameter Autoregressive Language Model. https://github.com/kingoflolz/mesh-transformer-jax.

Peifeng Wang, Aaron Chan, Filip Ilievski, Muhao Chen, and Xiang Ren. 2023. Pinto: Faithful language reasoning using prompt-generated rationales.

Yizhong Wang, Yeganeh Kordi, Swaroop Mishra, Alisa Liu, Noah A. Smith, Daniel Khashabi, and Hannaneh Hajishirzi. 2022. Self-instruct: Aligning language model with self generated instructions.

Jason Wei, Maarten Bosma, Vincent Y. Zhao, Kelvin Guu, Adams Wei Yu, Brian Lester, Nan Du, Andrew M. Dai, and Quoc V. Le. 2022. Finetuned language models are zero-shot learners.

Jason Wei, Xuezhi Wang, Dale Schuurmans, Maarten Bosma, Brian Ichter, Fei Xia, Ed Chi, Quoc Le, and Denny Zhou. 2023. Chain-of-thought prompting elicits reasoning in large language models.

Weihao Yu, Zihang Jiang, Yanfei Dong, and Jiashi Feng. 2020. Reclor: A reading comprehension dataset requiring logical reasoning. In *Proc. of ICLR*.

Eric Zelikman, Yuhuai Wu, Jesse Mu, and Noah D. Goodman. 2022. Star: Bootstrapping reasoning with reasoning.

Ruiqi Zhong, Kristy Lee, Zheng Zhang, and Dan Klein. 2021. Adapting language models for zero-shot learning by meta-tuning on dataset and prompt collections.

Yongchao Zhou, Andrei Ioan Muresanu, Ziwen Han, Keiran Paster, Silviu Pitis, Harris Chan, and Jimmy Ba. 2023. Large language models are human-level prompt engineers.

# A  Datasets Examples

We illustrate data examples mentioned in Section 3 here.

Figure 3: An instance from the EntailmentBank dataset.

Figure 4: An instance from the FOLIO dataset.

Figure 5: An instance from the ReClor dataset.

## B  The Instruction Types and Examples for the MRC Task

The instructions below further cultivate the model's critical thinking and argument analysis skills.

**Identify the Necessary Claim**: This instruction tasks the model to pinpoint the claim that must be true or is required for an argument to work. It is essentially training the model to identify essential assumptions or premises in an argument, thus honing its ability to understand argument structures.

**Strengthen an Argument**: Under this instruction, the model must identify information that would strengthen an argument. It requires the

model to not just understand the argument, but also anticipate what additional information could make the argument more convincing. This helps the model to improve its capability to enhance log-

| Instruction | Input | Output |
|---|---|---|
| **Translate the following inference to logic notation** | All squares have four sides. | ∀x (Square(x) → FourSides(x)) |
| **What can be inferred from the following premises in a single inference step** | David is working if and only if it is cloudy. David is not working. | If David works, then it is cloudy. If it is cloudy, then David is working. It is not cloudy. |
| **What can be inferred from the following premises in a single inference step? Name the inference rule being used** | For all x_9, x_9 is an artist if and only if x_9 will sleep. There is at least one x_9 for which x_9 is an artist if and only if x_9 will sleep. | For all x_9, if x_9 were an artist, then x_9 is sleeping can be inferred via the universal biconditional elimination rule. There is at least one x_9 for which if x_9 were an artist, then x_9 is sleeping can be inferred via the existential biconditional elimination rule. For all x_9, if x_9 sleeps, then x_9 is an artist can be inferred via the universal biconditional elimination rule. There is at least one x_9 for which if x_9 sleeps, then x_9 is an artist can be inferred via the existential biconditional elimination rule |
| **Consider the following premises ... Can we infer ... from them? name the inference rules being used** | Consider the following premises. If Charles takes a plane, then it is raining. If Patricia were curious, then Charles is taking a plane. If it rains, then Patricia is curious. If Charles takes a plane, then John is not reading a book. John is reading a book. Can we infer the following from them? If we can, name the inference rule being used. It is not raining | From the fact that if it rains, then Patricia is curious, and that if Patricia were curious, then Charles is taking a plane we can infer that if it rains, then Charles is taking a plane via transitivity. From the fact that if it rains, then Charles is taking a plane, and that if Charles takes a plane, then it is raining we can infer that it is raining if and only if Charles is taking a plane via biconditional introduction. From the fact that if Charles takes a plane, then John is not reading a book, and that John is reading a book we can infer that Charles is not taking a plane via modus tollens. Finally, from the fact that it is raining if and only if Charles is taking a plane, and that Charles is not taking a plane we can infer that it is not raining via biconditional elimination Therefore, the answer is yes. |

Table 7: The instruction types and illustrating examples for generating general inference CoT rationales.

ical arguments.

**Weaken an Argument**: This type is the opposite of Strengthen an Argument mentioned above. Here, the model is tasked with identifying information that would weaken an argument. This helps the model develop a nuanced understanding of argument structures and cultivate the ability to critique and dismantle arguments.

**Resolve a Situation**: This instruction requires the model to identify information that would explain or resolve a situation. This is about identifying missing information or finding potential solutions to a problem, further expanding the model's problem-solving capabilities.

**Identify a Flaw in Arguments Reasoning**: In this type, the model must identify a flaw in an argument's reasoning. This instruction cultivates the model's critical thinking skills, as it needs to scrutinize the argument and pinpoint any logical fallacies or inconsistencies.

By incorporating these instruction types, the data generation scheme is broadened to more complex logical reasoning tasks, particularly in the realm of argumentation and critical thinking, thereby enhancing the language model's ability to engage with more sophisticated and nuanced logical reasoning tasks.

## C Pseudo-Code for GPT-4 Generation

```
1: prompt_with_cot{
       'instruction': instruction,
       'input': input,
       'output': output,
       }
2: prompt_without_cot{
```

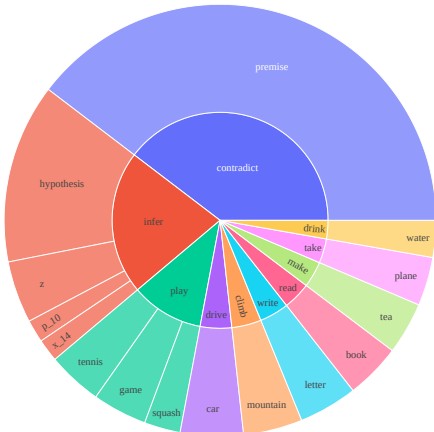

Figure 6: Root verb-noun pairs of GPT-4 generated responses. The inner circle of the plot contains the most frequent root verbs of GPT-4 outputs, and the outer circle displays the direct noun objects.

```
       'instruction': instruction,
       'input': input,
       }
3: output = openai.ChatCompletion.create(
       model = "gpt-4",
       messages = [{'role': 'system', 'content':
   instruction}, {'role': 'user', 'content': input},
   {'role': 'assistant', 'content': cot}]
       )
```

## D Data Statistics

## E Prompting Instruction-Tuned Models

LogiEval is an instruction-prompting style benchmark where each dataset instance has integrated instructions. This ensures the LLM comprehends the desired format and response requirements.

Below is the LogiEval instruction for the reading comprehension task:

| Instruction | Input | Output |
|---|---|---|
| identify the claim that must be true or is required in order for the argument to work. | Passage: Almost all of the books published in the past 150 years were printed on acidic paper. Unfortunately, every kind of acidic paper gradually destroys itself due to its very acidity. This process of deterioration can be slowed if the books are stored in a cool, dry environment. Techniques, which are now being developed, to deacidify books will probably be applied only to books with historical significance. Question: If all of the statements in the passage above are true, which one of the following must also be true? A. If a book was published in the past 150 years and is historically insignificant, it will probably deteriorate completely. B. Almost all of the books published in the past 150 years will gradually destroy themselves. C. Books published on acidic paper in 1900 should now all be at about the same state of deterioration. D. If a book is of historical significance and was printed before 150 years ago, it will be deacidified. Answer and reasoning step by step: | Answer: B. Almost all of the books published in the past 150 years will gradually destroy themselves. Reasoning: The passage states that almost all of the books published in the past 150 years were printed on acidic paper and that every kind of acidic paper gradually destroys itself due to its acidity. Therefore, it must be true that almost all of the books published in the past 150 years will gradually destroy themselves. |
| identify information that would strengthen an argument | Passage: The television show Henry was not widely watched until it was scheduled for Tuesday evenings immediately after That's Life, the most popular show on television. During the year after the move, Henry was consistently one of the ten most-watched shows on television. Since Henry's recent move to Wednesday evenings, however, it has been watched by far fewer people. We must conclude that Henry was widely watched before the move to Wednesday evenings because it followed That's Life and not because people especially liked it. Question: Which one of the following, if true, most strengthens the argument? A. The show that now follows That's Life on Tuesdays has double the number of viewers it had before being moved. B. Henry has been on the air for three years, but That's Life has been on the air for only two years. C. After its recent move to Wednesday, Henry was aired at the same time as the second most popular show on television. D. That's Life was not widely watched during the first year it was aired. Answer and reasoning step by step: | Answer: A. The show that now follows That's Life on Tuesdays has double the number of viewers it had before being moved. Reasoning: The argument suggests that Henry was widely watched not because people especially liked it, but because it followed That's Life, which was the most popular show on television. Option A strengthens this argument because it shows that another show that now follows That's Life on Tuesdays also experienced a significant increase in viewership after being moved to follow That's Life. This supports the idea that being scheduled after That's Life is the main factor contributing to a show's popularity, rather than the show's content or quality itself. |
| identify information that would weaken an argument | Passage: Several companies will soon offer personalized electronic news services, delivered via cable or telephone lines and displayed on a television. People using these services can view continually updated stories on those topics for which they subscribe. Since these services will provide people with the information they are looking for more quickly and efficiently than printed newspapers can, newspaper sales will decline drastically if these services become widely available. Question: Which one of the following, if true, most seriously weakens the argument? A. Approximately 30 percent of people have never relied on newspapers for information but instead have always relied on news programs broadcast on television and radio. B. In reading newspapers, most people not only look for stories on specific topics but also like to idly browse through headlines or pictures for amusing stories on unfamiliar or unusual topics. C. Companies offering personalized electronic news services will differ greatly in what they charge for access to their services, depending on how wide a range of topics they cover. D. The average monthly cost of subscribing to several channels on a personalized electronic news service will approximately equal the cost of a month's subscription to a newspaper. Answer and reasoning step by step: | B. In reading newspapers, most people not only look for stories on specific topics but also like to idly browse through headlines or pictures for amusing stories on unfamiliar or unusual topics. Reasoning: The argument is based on the assumption that these personalized electronic news services will replace newspapers because they provide information more quickly and efficiently. However, if most people also enjoy browsing through newspapers for amusing stories on unfamiliar or unusual topics (option B), then these electronic services may not provide the same experience for their users. This means that even if the electronic news services are more efficient in delivering specific news, they might not fully replace newspapers because people still enjoy the browsing experience that newspapers provide. This weakens the argument that newspaper sales will decline drastically if these services become widely available. |
| identify information that would explain or resolve a situation | Passage: On May first, in order to reduce the number of overdue books, a children's library instituted a policy of forgiving fines and giving bookmarks to children returning all of their overdue books. On July first there were twice as many overdue books as there had been on May first, although a record number of books had been returned during the interim. Question: Which of the following, if true, most helps to explain the apparent inconsistency in the results of the library's policy? A. The bookmarks became popular among the children, so in order to collect the bookmarks, many children borrowed many more books than they usually did and kept them past their due date. B. The children were allowed to borrow a maximum of five books for a two-week period, and hence each child could keep a maximum of fifteen books beyond their due date within a two-month period. C. Although the grace period enticed some children to return all of their overdue books, it did not convince all of the children with overdue books to return all of their books. D. Although the library forgave overdue fines during the grace period, the amount previously charged the children was minimal; hence, the forgiveness of the fines did not provide enough incentive for them to return their overdue books. Answer and reasoning step by step: | A. The bookmarks became popular among the children, so in order to collect the bookmarks, many children borrowed many more books than they usually did and kept them past their due date. Reasoning: The situation presents an inconsistency: despite forgiving fines and giving bookmarks as incentives to return overdue books, the number of overdue books doubled. We need to identify which statement helps explain this inconsistency. A. If the bookmarks became popular and children borrowed more books than usual to get them, it is possible that they couldn't read and return all those books on time, causing an increase in overdue books. This option explains the inconsistency, as the policy would have an unintended consequence - children borrowing too many books. B. This option only specifies the borrowing limit and duration, but it does not explain why the number of overdue books would increase despite the incentives. C. This option implies that not all children returned their overdue books, but it doesn't explain the doubling of the overdue books in the given period. It only suggests that the policy didn't work to its full potential. D. This option suggests that the fines were minimal, but it doesn't explain why the number of overdue books increased. If fines were minimal, we could assume that the situation would remain stable, not worsen. Therefore, option A most logically helps to explain the inconsistency in the results of the library's policy. |
| identify a flaw in an argument's reasoning | Passage: If you study history, then you will appreciate the vast differences among past civilizations, and you will appreciate these differences provided that you reflect on your own civilization. Hence, if you study history you will reflect on your own civilization. Question: Which one of the following is most closely parallel in its flawed reasoning to the argument above? A. One can become informed about the world provided that one reads the newspaper daily. If one is informed about the world, then one has an appreciation of other cultures. So if one reads the newspaper daily, then one can come to appreciate other cultures. B. If you learn Latin, you can improve your vocabulary, and you can improve your vocabulary if you study great works of literature. So you will study great works of literature if you learn Latin. C. Traveling to other countries deepens one's appreciation for their cultures, and this appreciation often encourages one to study the history of those lands. So the study of history increases one's desire to travel. D. By studying ancient art you begin to appreciate how much was accomplished with limited materials. Appreciation of ancient art leads to a deeper understanding of modern art. Therefore, studying ancient art can engender a profound new appreciation for modern art. Answer and reasoning step by step: | B. The flawed reasoning in the original argument is that it assumes that if one thing leads to another, then the reverse is also true. In the original argument, studying history is assumed to lead to reflecting on one's own civilization, and this flaw is parallel in option B, where learning Latin is assumed to lead to studying great works of literature. Both assume a reverse causation that isn't necessarily true. |

Table 8: The instruction types and examples of generating CoT rationales for multi-choice reading comprehension.

| Verb | Noun object | Count |
|---|---|---|
| consider | premise | 24388 |
| translate | inference | 13206 |
| answer | question | 1840 |

Table 9: Root verb-noun pairs of instructions.

Instructions: You will be presented with a passage and a question about that passage. There are four options to be chosen from, you need to choose the only correct option to answer that question. If the first option is right, you generate the answer 'A', if the second option is right, you generate the answer 'B', if the third option is right, you generate the answer 'C', if the fourth option is right, you generate the answer 'D', if the fifth option is right, you generate the answer 'E'. Read the question and options thoroughly and select the correct answer from the four answer labels. Read the passage thoroughly to ensure you know what the passage entails.

And the instructions for the NLI task:

Instructions: You will be presented with a set of facts and rules as premises, and a hypothesis about it. You need to decide whether the hypothesis is entailed by the premise by choosing one of the following answers: 'Yes': The hypothesis follows logically from the information contained in the premise. 'No': The hypothesis is logically false from the information contained in the premise. 'Neutral': It is not possible to determine whether the hypothesis is true or false without further information. Read the passage of information thoroughly and select the correct answer from the three answer labels. Read the premise thoroughly to ensure you know what the premise entails.

In the case of the MMLU dataset, we remain consistent with the protocols from previous evaluations. The prompt, "The following are multiple choice questions (with answers) about subject. question Answer:", provides both context and a clear response expectation.

## F  Ablation

## G  Case Study Example

| Dataset | LQ | LQ zh | RC | AL | LQ ood | CT | HL | TN | Overall |
|---|---|---|---|---|---|---|---|---|---|
| **Accuracy** | 43.83 | 28.94 | 45.40 | 19.82 | 29.57 | 33.09 | 31.47 | 53.00 | 35.64 |

Table 10: The performance on LogiEval by ablating "multi-choice". All results are in %.

**Passage:** A major health insurance company in Lagolia pays for special procedures prescribed by physicians only if the procedure is first approved as medically necessary by a company-appointed review panel. The rule is intended to save the company the money it might otherwise spend on medically unnecessary procedures. The company has recently announced that in order to reduce its costs, it will abandon this rule.
**Question:** Which of the following, if true, provides the strongest justification for the company's decision?
A.The review process is expensive and practically always results in approval of the prescribed procedure.
B. Patients often register dissatisfaction with physicians who prescribe nothing for their ailments.
C. The number of members of the company-appointed review panel who review a given procedure depends on the cost of the procedure.
D. Physicians often prescribe special procedures that are helpful but not altogether necessary for the health of the patient.
**Answer and reason:**

*A. The review process is expensive and practically always results in approval of the prescribed procedure.*

*Reason: This option provides a strong justification because if the review process is expensive and almost always approves the prescribed procedure, it makes no sense to continue with it, especially to save money. The company could simply approve the procedures without a review process, leading to higher costs. Therefore, the correct answer is A.*

Figure 7: An example where LLaMA-7b-logicot answers the question correctly. Model output is in *Italic*.

**Passage:** One seminar had 18 participants. It is known that :(1) At least 5 young teachers are female; (2) At least 6 female teachers are over middle age; (3) At least seven young women are teachers;
**Question:** According to the above information, which of the following can be concluded about the participants?
A. Some young teachers are not women
B. Some young women are not teachers
C. There are at least 11 young teachers
D. There are at least 13 female teachers
**Answer and reason:**

*(1) At least 5 young teachers are female - This means there is at least one young female teacher.*
*(2) At least 6 female teachers are over middle age - This means there are at least two female teachers over middle age.*
*(3) At least seven young women are teachers - This means there is at least two young women who are teachers.*
*(4) All young teachers are male or female - This is not possible, as we have two female teachers and two young women who are teachers.*
*(5) All female teachers are over middle age or they are not teachers - This means that all female teachers are over middle age or they are not teachers.*
*Now, considering that there is at least two young women who are teachers, we can infer the following:*
*(6) There is at least one young teacher who is a female - This is because there is at least one young women who is a teacher.*
*(11) Young teachers are either male or female - This is a direct result of the fact that there is at least one young teacher who is a female.*
*Now, we have established that there is at least one young female teacher.*
*(2) Young female teachers are either male or female - This statement is incorrect because we have already concluded that there is at least one young teacher who is a female.*
*(1) At least 5 young teachers are female - This is incorrect as there are only 5 young teachers in the seminar.*
*Thus, the correct answer is A.*

Figure 8: An example where LLaMA-7b-logicot answers the question incorrectly. Model output is in *Italic*.