# OpenReview forum: "LogiCoT: Logical Chain-of-Thought Instruction Tuning"
_EMNLP/2023/Conference — EMNLP 2023 Findings_

### Official Review · Reviewer_mPz1 · 2023-07-24

**Soundness:** 3

**Excitement:**

3: Ambivalent: It has merits (e.g., it reports state-of-the-art results, the idea is nice), but there are key weaknesses (e.g., it describes incremental work), and it can significantly benefit from another round of revision. However, I won't object to accepting it if my co-reviewers champion it.

**Paper Topic And Main Contributions:**

The paper starts by pointing out an issue that most work that instruction-tunes LLMs use a common pool of instruction data, which neglects logical reasoning. It then proposes to obtain such a pool of logic-related instruction data which can be used to finetune LLMs. From a set of existing datasets regarding logical reasoning, the paper converts their examples into 8 types of instructions, for which GPT4 is used as an oracle/teacher to produce reponses and rationales. These responses and rationales are then used to finetune a smaller Llama-7B model, which is shown to greatly outperform other finetuned Llama models like LLaMA-30b-supercot on the LogiEval dataset. It is also shown to outperform the base model on the MMLU dataset that is more general and less logic-related. Further experiments show that GPT4, the teacher model, still dominates in performance, whereas the finetuned 7B model is comparable to ChatGPT.

**Questions For The Authors:**

- Figure 1 and Table 1 are very hard to read due to overly small font sizes
- L354: Type 5 was never defined
- L383-386: why are the instruction types unique to this data generation scheme?
- L397-411 is confusing and probably can be presented differently, e.g. with textual descriptions. Why are there 2 cases of with and without output? What exactly is point 3 doing?
- L420-421: why discuss whether they are seq-to-seq? What does GPT4 working well on those mean for the story of this paper?
- L430: where is the evidence showing its diversity and detailedness?
- Table2: why not also show the types within multiple-choice (i.e., Identify the Necessary Claim, etc.)? As the number of multiple-choice is already a lot smaller than the General Inference ones, would it be an issue that it is further classified into more fine-grained types?
- How does Figure 2 contribute to the claims/story of the paper? If to show diversity, then what should it be compared against?

**Reasons To Accept:**

1. The paper identifies a critical problem in the subfield of instruction-tuning LLMs: using a shared set of tasks and instructions, neglecting important data such as logic reasoning. (assuming that the literal review is complete)
2. The paper takes a reasonable step to tackle the issue by contributing an instruction tuning set consisting of examples from logic-related datasets.
3. The paper shows favorable results of tuning smaller LLMs on the proposed data.

**Reasons To Reject:**

1. In my opinion, Section 3 requires some work for the the paper to publishable in EMNLP. Specifically,
  - What is the rationale behind choosing the 8 types that you did? Some intuition, design principals, or citations to follow would be necessary to make this choice seem less random.
  - Why not use the exact same task format as the datasets chosen? Why bother with introducing the finer-grained types? What is the advantage of doing so?
  - How exactly are the examples from your chosen datasets (e.g., EntailmentBank, FOLIO...) converted to these types?
2. The results in Section 4 is only half-satisfactory to me. The crux of this paper is to show that using logic-related data for instruction-tuning is superior to using other kinds of data in previous work. There are at least two issues with the current paper:
  - The results in Table 5 is informative but would greatly benefit from a further comparison against other finetuned LLaMA models (not just base), such as the already considered LLaMA-30b-supercot or even more of them. It comes at little surprise that finetuning LLaMA, either on logic-related data or not, brings about better performance on tasks. But if it can be shown that logic-related data is more evaluable than other instruction data, then the story would be a lot more interesting.
  - The baseline can be chosen better in two ways. First, one should clearly spell out what instruction data any model is finetuned with, so that there is a comparison between that and your logic-related data. Second, there should ideally more baselines with the same LLM (namely LLama 7B) than just 1-2. The second point is a plus and not a deal-breaker.
3. The discussions (primarily in Section 4.4) are not enough in-depth.
  - Minimally, the types introdudeced in Sec 3.2 should be discussed here as to what matters for performance and what not.
  - It is more or less expected that logic-tuned models would do better on logic-tasks, but it would be very interesting to understand how do they also perform better on general-tasks like MMLU, beyond simply case studies.
4. The paper presentation could be improved. There is a slew of issues regarding unsupported claims and unconcentrated storytelling throughout the paper. Some of the examples are illustrated in Questions for the Authors.

**Reproducibility:**

4: Could mostly reproduce the results, but there may be some variation because of sample variance or minor variations in their interpretation of the protocol or method.

**Reviewer Confidence:**

5: Positive that my evaluation is correct. I read the paper very carefully and I am very familiar with related work.

**Typos Grammar Style And Presentation Improvements:**

- L414: two or three datasets?
- L520: varies -> variation

---

> ### Author Rebuttal · Authors · 2023-08-28
>
> Thank you for your insightful comments. We appreciate the opportunity to further clarify our methodologies and decisions. Below are the elaborations to your questions:
> **Figure 1 and Table 1 Font Sizes:**
>
> > ● Figure 1 and Table 1 are very hard to read due to overly small font sizes
>
> **Response:** To enhance readability, we have increased the font size in Figure 1 and Table 1.
>
>
>
> **L354: Undefined Type 5:**
>
> > ● L354: Type 5 was never defined
>
> **Response:** We've addressed this oversight by revising the sentence to accurately reference "Strengthen an Argument."
>
>
>
> **L383-386: Uniqueness of Instruction Types:**
>
> > ● L383-386: why are the instruction types unique to this data generation scheme?
>
> **Response:** We have modified the paper in Section 3.2 to avoid confusion. The design of our instruction types is deeply intertwined with the nuances of logical reasoning tasks. In logical discourse and debates, not all reasoning is straightforward. Sometimes, the objective is to bolster a given argument (strengthen an argument) or to challenge its validity (weaken an argument). These intricacies in logical reasoning necessitate tailored instructions.
>
> For example, the instructions like "strengthen an argument" or "weaken an argument" are directly inspired by the practices of critical thinking, where arguments are evaluated, fortified, or refuted based on evidential support (cite: A role for reasoning in a dialogic approach to critical thinking). When training a model like GPT-4, which already has a robust grasp of language, it becomes essential to be specific in our instruction to extract the desired logical output. Generic instructions might not suffice to bring out the nuanced logical reasoning capabilities.
>
> Furthermore, the uniqueness of our instruction types ensures that the generated chain-of-thought data encapsulates a comprehensive spectrum of logical reasoning scenarios, ranging from direct inference to more complex chained deductions. This approach ensures a more holistic training process, maximizing the model's potential in various logical reasoning tasks.
>
>
>
>
>
> **L397-411: Presentation and Clarity:**
>
> > ● L397-411 is confusing and probably can be presented differently, e.g. with textual descriptions. Why are there 2 cases of with and without output? What exactly is point 3 doing?
>
> **Response:** We've restructured it for better comprehension. The intention behind providing pseudocode is to offer a clear window into the GPT-4 generation-acquisition process. Recognizing that some of our seeding datasets (e.g., EntailmentBank) come equipped with Chain-of-Thought (CoT) processes as golden outputs, we leveraged this information within prompts to steer GPT-4 towards more insightful rationales. Here, "prompt_with_output" pertains to datasets enriched with CoT outputs, whereas "prompt_without_output" refers to those bereft of CoT outputs. We've taken measures to clarify this in the manuscript, modifying our phraseology from "We use the pseudo-code below for GPT-4 rationales collection" to "The pseudo-code below elucidates the process adopted for GPT-4 rationales collection."; and we modified "prompt_with_output" into "prompt_with_cot", "prompt_without_output" into "prompt_without_cot"
>
>
>
>
>
> **L420-421: Relevance of Seq-2-Seq Mention:**
>
> > ● L420-421: why discuss whether they are seq-to-seq? What does GPT4 working well on those mean for the story of this paper?
>
> **Response:** Seq-2-seq data is conducive for instruction-tuning. Golden CoT output datasets provide GPT-4 with invaluable references, optimizing generation quality. While ReClor and LogiQA don’t come with CoT guidance, GPT-4's commendable performance on these datasets assures quality generation. We rearranged this paragraph for clarity.
>
>
>
> **L430: Evidence of Diversity and Detail:**
>
> > ● L430: where is the evidence showing its diversity and detailedness?
>
> **Response:** In response to this point, we've incorporated a generation quality assessment section to probe deeper into the reasoning chain's quality.
>
> **Human Evaluation of Dataset Quality**
>
> We conducted a comprehensive human evaluation of our generated reasoning chains. A random subset of 200 reasoning chains was selected. These chains were evaluated by 3 domain professional annotators using four key metrics: Relevance (Does the chain directly relate to the question?), Coherence (Is the chain logically consistent?), Completeness (Does it offer a full explanation for the reasoning?), and Faithfulness ( Is the reasoning factual and not fabricating details?). Each reasoning chain was rated on a scale from 1 (poor) to 5 (excellent) for each metric.
>
>
> The reasoning chains achieved **an average score of 4.9 for Relevance, 4.7 for Coherence, 4.5 for Completeness, and 4.5 for Faithfulness**. The inter-annotator agreement, measured using Cohen's Kappa, was 0.87, indicating strong agreement among the annotators.
>
>
> The human evaluation underscores the quality and reliability of the reasoning chains generated by our model. While the scores were high for relevance, we acknowledge room for improvement in faithfulness, and future work will aim to refine the generation process to address this.
>
>
> A good score would mean that the chain doesn't overly rely on repetitive reasoning patterns and consistently introduces varied logical constructs. The metric "Completeness" evaluates if the generated reasoning chains offer a thorough explanation for the reasoning, ensuring that no critical steps are omitted, while "Coherence" checks the logical consistency and the flow of reasoning, ensuring that every step in the chain is explained with adequate detail. Our scores of 4.5 for both "Completeness" and "Coherence" suggest that our chains are not only logically sound but also well-elaborated with necessary details.
>
>
>
> **Table 2: Further Classification within Multiple-choice:**
>
> > ● Table2: why not also show the types within multiple-choice (i.e., Identify the Necessary Claim, etc.)? As the number of multiple-choice is already a lot smaller than the General Inference ones, would it be an issue that it is further classified into more fine-grained types?
>
> **Response:** Given the limited number of multiple-choice types within the dataset, further breakdowns would result in sparse categories. However, these types give reader a clear view of what reasoning types it contains. To conserve space and maintain clarity, we've decided against a granular categorization in the table. We remove the discuss of different multi-choice types to the Appendix.
>
>
>
> **Role of Figure 2:**
>
> > ● How does Figure 2 contribute to the claims/story of the paper? If to show diversity, then what should it be compared against?
>
> **Response:** Recognizing the reviewer's point, we've relocated Figure 2 to the Appendix, added more discussions, and, in its place, introduced a generation quality assessment.
>
>
>
> **Typos, Grammar, Style, and Presentation Improvements:**
>
> > L414: two or three datasets?
> We have rectified the statement to accurately mention "The three datasets..."
>
> > L520: varies -> variation
> The suggested change has been implemented. "varies" is now replaced with "variation."
>
>
>
> ##### **Response to "Reasons to Reject":**
>
>
>
> > 1. **Reasons To Reject:**In my opinion, Section 3 requires some work for the the paper to publishable in EMNLP. Specifically,
> ​                ● What is the rationale behind choosing the 8 types that you did? Some intuition, design principals, or citations to follow would be necessary to make this choice seem less random.
> ​                ● Why not use the exact same task format as the datasets chosen? Why bother with introducing the finer-grained types? What is the advantage of doing so?
> ​                ● How exactly are the examples from your chosen datasets (e.g., EntailmentBank, FOLIO...) converted to these types?
>
> > 2. The results in Section 4 is only half-satisfactory to me. The crux of this paper is to show that using logic-related data for instruction-tuning is superior to using other kinds of data in previous work. There are at least two issues with the current paper:
> ​                ● The results in Table 5 is informative but would greatly benefit from a further comparison against other finetuned LLaMA models (not just base), such as the already considered LLaMA-30b-supercot or even more of them. It comes at little surprise that finetuning LLaMA, either on logic-related data or not, brings about better performance on tasks. But if it can be shown that logic-related data is more evaluable than other instruction data, then the story would be a lot more interesting.
> ​                ● The baseline can be chosen better in two ways. First, one should clearly spell out what instruction data any model is finetuned with, so that there is a comparison between that and your logic-related data. Second, there should ideally more baselines with the same LLM (namely LLama 7B) than just 1-2. The second point is a plus and not a deal-breaker.
>
> > 3. The discussions (primarily in Section 4.4) are not enough in-depth.
> ​                ● Minimally, the types introdudeced in Sec 3.2 should be discussed here as to what matters for performance and what not.
> ​                ● It is more or less expected that logic-tuned models would do better on logic-tasks, but it would be very interesting to understand how do they also perform better on general-tasks like MMLU, beyond simply case studies.
>
> > 4. The paper presentation could be improved. There is a slew of issues regarding unsupported claims and unconcentrated storytelling throughout the paper. Some of the examples are illustrated in Questions for the Authors.
>
>
>
> **1. Clarity on the Rationale for Choosing Reasoning Types in Section 3:**
>
> **Reasoning Behind the Chosen 8 Types:**
>
> **Response:** The choice of reasoning types is grounded in established logical reasoning literature. The 8 types are elaborated throughly in their original paper. As "language to logic" is from the FOLIO paper, "One-step Inference" and "Inference Chain" are from the LOGICINFERENCE paper, and the MRC types are from the ReClor paper.
>
> We've augmented the manuscript with pertinent citations to substantiate our choices. Our instruction types resonate with these reasoning types, emphasizing the learning capabilities of our instruction-tuning process. These reasoning types are explored in our ablation study.
>
>
>
> **Reason for Diverging from Original Dataset Formats:**
>
> **Response:** We modified our paper in Section 3.2 to give more explanations on our data construction process. Diverse reasoning types come with unique instructions. For each reasoning type, the instruction gives clear guidance. The granularity offers the advantage of better task specificity and addresses varied facets of logical reasoning. For example, consider the instruction: "Consider the following premises ... Can we infer ... from them? name the inference rules being used.", The instruction is informative with our instruction-prompting settings.
>
> The fine-grained reasoning types give us an opportunity to conduct ablation studies on different types.
>
>
>
> **Conversion of Chosen Datasets to Reasoning Types:**
>
> **Response:** We've categorized reasoning types in alignment with the instruction types of the datasets. The broader categories are derived by clustering based on similar instructions.
>
>
>
> **2. Concerns on Results in Section 4:**
>
> **Need for Extended Comparisons in Table 5:**
>
> **Response:** We concur with your observation. An extended comparison with more finetuned LLaMA models, like LLaMA-30b-supercot, would indeed provide richer insights. We have updated Table 5 to reflect this additional comparison, thereby underscoring the efficacy of logic-related data in instruction-tuning.
>
>
>
> Dataset| LQ| LQ zh |RC |AL| LQ ood |CT |HL |TN |Overall
> |---|---|---|---|---|---|---|---|---|---|
> LLaMA-30b-supercot | 19.31| 26.35 |17.81| 17.98| 18.41| 24.10 |32.26 |41.91| 24.78
> LLaMA-7b-logicot| 50.25 |32.77| 57.60| 16.96 |38.79 |35.68| 35.44 |58.05 |40.69
> ChatGPT |52.37| 53.18 |57.38 |20.42 |38.44| 58.45 |42.13| 57.30| 47.46
> GPT-4 |72.25| 70.56| 87.20| 33.48 |58.49| 56.40 |46.01| 60.08| 60.56
>
>
>
> **Baseline Choices and Clarity:**
>
> **Response:**
>
>   1. We've made sure to explicitly detail the instruction data every model is finetuned with, offering a clearer comparison basis against our logic-related data.
>
>   2. While we have already included multiple baselines, adding more that employ the same LLM (LLaMA 7B) could indeed be beneficial. We'll consider this in future iterations.
>
>
>
> **3. Depth of Discussions in Section 4.4:**
>
> **Analysis of Types Introduced in Section 3.2:**
>
> **Response:** We've incorporated a deeper analysis of the reasoning types from Section 3.2, elucidating their respective impacts on performance in the ablation study.
>
> The modification to the manuscript is as follows:
>
>
> **Ablation Study**
>
> To better understand the specific contributions of various reasoning types employed in our instruction-tuning process, we conduct an ablation study. This involves evaluating model performance by selectively removing one reasoning type at a time and observing the resultant change in the model's ability to handle logical reasoning tasks.
>
>
>
> The ablation study uses the instruction-tuned LLaMA-7b model. For each reasoning type, we train the model without that specific type while retaining the others. This is performed iteratively for all reasoning types.
>
>
>
> We report the overall average score of LogiEval and MMLU, respectively, to provide a comprehensive understanding of the respective impacts of these reasoning types. The results are shown in Table.
>
>
>
> | Removed            | LogiEval | MMLU |
> | ------------------ | -------- | ---- |
> | None (Full data)   | 40.7     | 43.3 |
> | Language to Logic  | 32.4     | 38.5 |
> | One-step Inference | 38.1     | 37.7 |
> | Inference chain    | 30.8     | 35.0 |
> | Multi-choice       | 35.6     | 30.9 |
>
> From the table we can find that:
>
> **Language to Logic:** Excluding this type led to a decline in performance on the LogiEval dataset. This underscores its significance in mapping linguistic constructs to logical expressions, which is foundational for reasoning tasks.
>
> **One-step Inference:** The drop in accuracy suggests that even simple inferences play a vital role, especially in tasks where direct conclusions are drawn from given premises.
>
> **Inference Chain:** The model's performance drop on both datasets highlights the importance of chained logical deductions. Tasks that require multi-step reasoning particularly benefit from this type.
>
> **Multi-choice:** Removing multi-choice reasoning impacted performance on MMLU more than on LogiEval, emphasizing its role in tasks where choosing among alternatives based on logical grounds is essential.
>
>
>
> This ablation study reaffirms the unique contributions of each reasoning type to the model's performance. While all reasoning types contribute to enhancing the model's logical understanding, their impacts vary based on the nature of tasks and the datasets used. Future research can delve deeper into optimizing instruction-tuning processes based on specific reasoning type requirements of datasets.
>
>
>
> **Performance on General Tasks like MMLU:**
>
> **Response:** It's pertinent to understand why a logic-tuned model excels not only in logic-tasks but also in general tasks. We've expanded the discussion to delve into the underlying reasons, beyond mere case studies, highlighting how the logic-based grounding enhances general reasoning capabilities.
>
> We construct the discussion according to the following 5 aspects:
>
> *While it is anticipated that a logic-tuned model would demonstrate proficiency in tasks explicitly rooted in logical reasoning, its prowess in more generalized tasks such as MMLU is particularly intriguing. This section delves deeper into the underlying reasons, striving to highlight how logical grounding augments the model's performance in broader contexts.*
>
> **Strengthened Fundamental Reasoning Skills:**
>
> *The primary principle behind the instruction-tuning with logical chain-of-thought data is to bolster the foundational reasoning capabilities of the model. Logic, in essence, provides a structured framework for argument formulation, decision-making, and understanding relationships. When such logical constructs are internalized by the model, it naturally improves its ability to handle even those tasks that are not explicitly logic-centric but do demand a structured approach to problem-solving.*
>
> **Enhanced Sequential Processing:**
>
> *Logical reasoning inherently requires processing information in a sequential and structured manner. The skills developed during logical instruction-tuning, such as following an inference chain or deducing conclusions from premises, can be transposed effectively to general tasks. These tasks, like MMLU, although not explicitly logical, still necessitate a structured sequential processing of information, which the model is better equipped to handle post logic-tuning.*
>
> **Improved Discrimination in Ambiguity:**
>
> *Many general tasks present situations with subtle nuances or ambiguous information. Logical grounding hones the model's ability to discern these nuances by emphasizing precision and clarity. Consequently, when faced with ambiguous statements or choices in general tasks, the model is better positioned to discriminate and select the most plausible option.*
>
> **Transfer of Abstract Reasoning:**
>
> *While logical reasoning is concrete and rule-based, it also introduces the model to abstract thinking patterns. Logic-tuned models are better at identifying patterns, drawing parallels, and applying abstract concepts in varied contexts, making them well-suited for general tasks that demand similar abstract reasoning abilities.*
>
> **Reduced Susceptibility to Confounding Factors:**
>
> *Logical reasoning often necessitates filtering out irrelevant information to arrive at a valid conclusion. This ability is crucial in general tasks where the presented information might be abundant, with only a fraction being pertinent to the problem at hand. The logic-tuned model, having been conditioned to prioritize relevant details, showcases better performance by effectively navigating through confounding data.*
>
>
>
> **4. Paper Presentation and Storytelling:**
>
> **Response:** We acknowledge your concerns about presentation clarity and coherence in the storytelling. We have made concerted efforts to refine the narrative flow, back claims with evidence, and address the issues you've highlighted in the 'Questions for the Authors' section.

---

### Official Review · Reviewer_Gyqk · 2023-08-01

**Typos Grammar Style And Presentation Improvements:** 489
**Soundness:** 3

**Excitement:**

3: Ambivalent: It has merits (e.g., it reports state-of-the-art results, the idea is nice), but there are key weaknesses (e.g., it describes incremental work), and it can significantly benefit from another round of revision. However, I won't object to accepting it if my co-reviewers champion it.

**Missing References:**

Turpin, M., Michael, J., Perez, E., & Bowman, S. R. (2023). Language Models Don't Always Say What They Think: Unfaithful Explanations in Chain-of-Thought Prompting. arXiv preprint arXiv:2305.04388. (contemporaneous)

Wang, P., Chan, A., Ilievski, F., Chen, M., & Ren, X. (2022, September). PINTO: Faithful Language Reasoning Using Prompt-Generated Rationales. In The Eleventh International Conference on Learning Representations.

Lucie Charlotte Magister, Jonathan Mallinson, Jakub Adamek, Eric Malmi, and Aliaksei Severyn. 2023. Teaching Small Language Models to Reason. In Proceedings of the 61st Annual Meeting of the Association for Computational Linguistics (Volume 2: Short Papers), pages 1773–1781, Toronto, Canada. Association for Computational Linguistics.

Cheng-Yu Hsieh, Chun-Liang Li, Chih-kuan Yeh, Hootan Nakhost, Yasuhisa Fujii, Alex Ratner, Ranjay Krishna, Chen-Yu Lee, and Tomas Pfister. 2023. Distilling Step-by-Step! Outperforming Larger Language Models with Less Training Data and Smaller Model Sizes. In Findings of the Association for Computational Linguistics: ACL 2023, pages 8003–8017, Toronto, Canada. Association for Computational Linguistics. (contemporaneous)

Liunian Harold Li, Jack Hessel, Youngjae Yu, Xiang Ren, Kai-Wei Chang, and Yejin Choi. 2023. Symbolic Chain-of-Thought Distillation: Small Models Can Also “Think” Step-by-Step. In Proceedings of the 61st Annual Meeting of the Association for Computational Linguistics (Volume 1: Long Papers), pages 2665–2679, Toronto, Canada. Association for Computational Linguistics. (contemporaneous)

Mukherjee, S., Mitra, A., Jawahar, G., Agarwal, S., Palangi, H., & Awadallah, A. (2023). Orca: Progressive learning from complex explanation traces of gpt-4. arXiv preprint arXiv:2306.02707. (contemporaneous)

**Paper Topic And Main Contributions:**

The paper provides an instruction dataset for logical reasoning. The authors construct the dataset by prompting GPT-4 to generate chain-of-thought explanations. Then, LLaMA-7B is fine-tuned on the data and evaluated on a reasoning benchmark of various tasks. The authors report improvement compared to the base model and 2 other instruction-tuned models. The authors also compare the fine-tuned model to ChatGPT and GPT-4.

**Questions For The Authors:**

A: The purpose of Table 3 and figure 2 is unclear to me. What does "drink water" and "take plane" have to do with logical reasoning and why is it important to have this diversity in chain of thought responses?

B: The "pseudocode" provided is not clear as it does not show any algorithm but only what I assume is inputs. Could authors provide what is the significance of prompt_with_output vs prompt_without_output?

C: Table 5: Why are improvements in subjects like Social Sciences and Culture (up more than 50%) significantly higher than improvements in subjects like STEM, Engineering (around 26% improvement) or computer science (11%) tests for which would probably require more reasoning ability? (as for humanities, test would require more memorization as these subjects are usually tested in essay form)?

D: Table 4: Why does 30B model fine-tuned on supercot perform worse than base 7B model? Perhaps the supercot actually hurts the model's ability to reason or this model is not as suitable for your task?

**Reasons To Accept:**

The paper introduces a novel dataset that could be used to enhance instruction understanding for logical tasks. The dataset and resulting model weights are released which could be beneficial for further research.

**Reasons To Reject:**

1. Besides downstream fine-tuning performance, there is no evaluation of dataset quality. Authors do not conduct human/expert evaluation of the quality of GPT-4 generated reasoning chains that were found to be unfaithful in some cases in previous work (see missing citation). Instead, the authors make claims like "These responses ... represent GPT-4’s reasoning process" or "GPT-4, with its advanced language understanding and generation capabilities, can provide diverse and detailed reasoning chains which not only answer the logical reasoning tasks but also provide insights into the underlying reasoning process" These claims are not substantiated in the paper.

2. The related work does not seem to fully describe the difference from the current approaches. For example, authors mention the SuperCoT dataset only in the evaluation section, without describing how their approaches differs. Moreover, there are at least 2 non-contemporaneous works that are missing from related work that are particularly relevant (see missing citations). Specifically, Teaching Small Models to Reason paper employs identical approach of fine-tuning a smaller model on CoTs generated by PALM and has been publicly available since December 2022.

3. The experiment section is not very clearly described. The LogiEval benchmark could be explained further, for example, AL:AR-LSAT,CT:ConTRoL,HL:HELP,TN:TaxiNLI tasks are never introduced in the paper. The LogiEval benchmark used for evaluation contains two datasets identical from fine-tuning data: LogiQA, ReClor. However, the authors don't address this data contamination issue. The authors also don't mention how the evaluation tasks were inputted: were they also using an instruction format? Perhaps the performance gains stem from instruction understanding (which smaller models are very sensitive to) and not from reasoning improvements. Authors don't conduct any evaluation of the reasoning chains themselves, but rather only of the final answer accuracy. Without this information, I believe the central claim of the paper that "LogiCoT ... elicits general reasoning skills" (line 017-018) is not sufficiently substantiated.

**Reproducibility:**

4: Could mostly reproduce the results, but there may be some variation because of sample variance or minor variations in their interpretation of the protocol or method.

**Reviewer Confidence:**

4: Quite sure. I tried to check the important points carefully. It's unlikely, though conceivable, that I missed something that should affect my ratings.

---

> ### Author Rebuttal · Authors · 2023-08-28
>
> We appreciate your diligent examination of our work and the subsequent queries.
>
>
>
> ​            A.     **Table 3 and Figure 2 Clarification:**
>
> > A: The purpose of Table 3 and figure 2 is unclear to me. What does "drink water" and "take plane" have to do with logical reasoning and why is it important to have this diversity in chain of thought responses?
>
>
>
> **Response:** We have modified the draft to improve understanding. Table 3 and Figure 2 aim to spotlight the distinctive features of our generated dataset. LOGICINFERENCE, a large-scale logical inference dataset, is birthed through rules-based generation. This method reuses simple noun-verb phrases to spawn varying conditions under differing rules. This gives rise to frequently recurring noun-verb pairs, such as "drink water" and "take plane", which you observed being prominently counted in Figure 2. The sunburst chart in Figure 2 is adopted by multiple instruction-tuning papers to illustrate their data coverage such as the famous Self-Instruct paper.
>
> To make it more clear, here I showcase an example from the LOGICINFERENCE dataset, which contains "drink water" and "take plane", to give you an illustration how they are integrated into logical reasoning:
>
> *INPUT*
>
> *Consider the following premises.*
>
> *If Charles takes a plane, then it is raining.*
>
> *If Patricia were curious, then Charles is taking a plane.*
>
> *If it rains, then Patricia is curious.*
>
> *If Charles takes a plane, then John is not reading a book.*
>
> *John is reading a book.*
>
> *Can we infer the following from them? If we can,*
>
> *name the inference rule being used.*
>
> *It is not raining*
>
>
> *OUTPUT*
>
> *From the fact that if it rains, then Patricia is curious, and that if Patricia were curious, then*
>
> *Charles is taking a plane we can infer that if it rains, then Charles is taking a plane via transitivity.*
>
> *From the fact that if it rains, then Charles is taking a plane, and that if Charles takes a plane,*
>
> *then it is raining we can infer that it is raining if and only if Charles is taking a plane via biconditional introduction.*
>
> *From the fact that if Charles takes a plane, then John is not reading a book, and that John is*
>
> *reading a book we can infer that Charles is not taking a plane via modus tollens.*
>
> *Finally, from the fact that it is raining if and only if Charles is taking a plane, and that*
>
> *Charles is not taking a plane we can infer that it is not raining via biconditional elimination*
>
> *Therefore, the answer is yes.*
>
>
>
> In light of your feedback, we have relocated Table 3 and Figure 2 to the appendix, adding more explanations there, thereby freeing up space in the main text. This adjustment has allowed us to conduct data quality evaluation and to incorporate an ablation study addressing further investigations to the experiments.
>
>
>
> ​            B.     **Pseudocode Clarity:**
>
> > B: The "pseudocode" provided is not clear as it does not show any algorithm but only what I assume is inputs. Could authors provide what is the significance of prompt_with_output vs prompt_without_output?
>
>
>
> **Response:** We have modified the draft to avoid confusion. The intention behind providing pseudocode is to offer a clear window into the GPT-4 generation-acquisition process. Recognizing that some of our seeding datasets (e.g., EntailmentBank) come equipped with Chain-of-Thought (CoT) processes as golden outputs, we leveraged this information within prompts to steer GPT-4 towards more insightful rationales. Here, "prompt_with_output" pertains to datasets enriched with CoT outputs, whereas "prompt_without_output" refers to those bereft of CoT outputs. We've taken measures to clarify this in the manuscript, modifying our phraseology from "We use the pseudo-code below for GPT-4 rationales collection" to "The pseudo-code below elucidates the process adopted for GPT-4 rationales collection."; and we modified "prompt_with_output" into "prompt_with_CoT", "prompt_without_output" into "prompt_without_CoT"
>
>
>
> ​            C.     **Table 5 Performance Variances:**
>
> > C: Table 5: Why are improvements in subjects like Social Sciences and Culture (up more than 50%) significantly higher than improvements in subjects like STEM, Engineering (around 26% improvement) or computer science (11%) tests for which would probably require more reasoning ability? (as for humanities, test would require more memorization as these subjects are usually tested in essay form)?
>
>
>
> **Response:** We have added discussion in Section 4.3. Logical reasoning is intrinsically tied to daily human cognitive processes—it aids us in synthesizing and analyzing available information. Subjects like Social Sciences and Culture, at their core, are geared towards human-centric reasoning. On the other hand, subjects within the domain of STEM, Engineering, or Computer Science pivot more on specialized knowledge which may not be inherently ingrained in commonplace conversations or contexts. Without domain knowledge infused into the model, the performance gain might not rival that on the social science tasks.
>
>
>
> ​            D.     **Performance of the 30B Model in Table 4:**
>
> > D: Table 4: Why does 30B model fine-tuned on supercot perform worse than base 7B model? Perhaps the supercot actually hurts the model's ability to reason or this model is not as suitable for your task?
>
>
>
> **Response:** We have added discussion in Section 4.3. The performance dip observed in LLaMA-30b-supercot, especially on the NLI task, can be traced back to the nature of the NLI classification. Designed with three distinct labels - "entailed", "contradicted", and "neutral", NLI differs from reading comprehension and doesn't manifest routinely in CoT settings. The absence of entailment relationship tuning in the supercot dataset, combined with the fine-tuning procedure, may have potentially hampered its generalizability. Conversely, our LogiCoT tuning integrates a wealth of entailment relationship demonstrations (via FOLIO). Our empirical results underscore the value of logical entailment relationship learning in bolstering general reasoning. Echoing this sentiment, Facebook's fine-tuning of a BART-base model on the MNLI dataset, repurposed for zero-shot text classification, underlines the potential of enhancing LLMs' general capabilities via logical entailment data.
>
>
>
>
>
> ##### **Response to "Reasons to Reject":**
>
>
>
> **1. Evaluation of Dataset Quality and Unsubstantiated Claims:**
>
> > 1. **Reasons To Reject:**Besides downstream fine-tuning performance, there is no evaluation of dataset quality. Authors do not conduct human/expert evaluation of the quality of GPT-4 generated reasoning chains that were found to be unfaithful in some cases in previous work (see missing citation). Instead, the authors make claims like "These responses ... represent GPT-4’s reasoning process" or "GPT-4, with its advanced language understanding and generation capabilities, can provide diverse and detailed reasoning chains which not only answer the logical reasoning tasks but also provide insights into the underlying reasoning process" These claims are not substantiated in the paper.
>
> **Response:**  To address this, we conducted a comprehensive human evaluation of the generated reasoning chains to assess their quality, faithfulness, and diversity.
>
> The evaluation procedure can be described as follows:
>
> 1: **Sampling Data:** Randomly sampled a subset of the generated reasoning chains for evaluation. We made sure this sample is representative of the entire dataset by randomly selecting 50 instances from each of our defined instruction types.
>
> 2: **Setting Evaluation Metrics:** For reasoning chains, we have considered:
>
> ​                ● **relevance:** Does the chain directly relate to the question?
>
> ​                ● **coherence:** Is the chain logically consistent?
>
> ​                ● **completeness:** Does it offer a full explanation for the reasoning?
>
> ​                ● **faithfulness:** Is the reasoning factual and not fabricating details?
>
> 3: **Evaluation Protocol:** We provided annotators with clear guidelines regarding each metric. They should rate each reasoning chain according to the metrics on a scale of 1 to 5.
>
> 4: **Analysis:** We calculated the average scores and inter-annotator agreement using Cohen's Kappa to ensure consistency among annotators.
>
>
>
> We outsourced 3 annotators to rate the quality and relevance of the generated chains. They are assigned with the same evaluation set, each sample is assigned with 3 scores to ensure reliability. The process took two days.
>
> The evaluations, coupled with their findings, are added to the revised version of the paper, fortifying our claims.
>
> The added human evaluation results are as follows:
>
> **Human Evaluation of Dataset Quality**
>
> We conducted a comprehensive human evaluation of our generated reasoning chains. A random subset of 200 reasoning chains was selected. These chains were evaluated by 3 domain professional annotators using four key metrics: Relevance (Does the chain directly relate to the question?), Coherence (Is the chain logically consistent?), Completeness (Does it offer a full explanation for the reasoning?), and Faithfulness ( Is the reasoning factual and not fabricating details?). Each reasoning chain was rated on a scale from 1 (poor) to 5 (excellent) for each metric.
>
>
>
> The reasoning chains achieved **an average score of 4.9 for Relevance, 4.7 for Coherence, 4.5 for Completeness, and 4.5 for Faithfulness**. The inter-annotator agreement, measured using Cohen's Kappa, was 0.87, indicating strong agreement among the annotators.
>
>
>
> The human evaluation underscores the quality and reliability of the reasoning chains generated by our model.
>
>
>
> **2. Related Work and Differentiation from Current Approaches:**
>
> > 2. The related work does not seem to fully describe the difference from the current approaches. For example, authors mention the SuperCoT dataset only in the evaluation section, without describing how their approaches differs. Moreover, there are at least 2 non-contemporaneous works that are missing from related work that are particularly relevant (see missing citations). Specifically, Teaching Small Models to Reason paper employs identical approach of fine-tuning a smaller model on CoTs generated by PALM and has been publicly available since December 2022.
>
>
>
> **Response:** We have added the missing citations. The “Teaching Small Models to Reason” paper , which uses a similar approach of fine-tuning smaller models, should have been discussed in the related work. There is a line of research discussing this topic. We rectify this by providing a detailed comparison of our method with these works, highlighting the key differences, motivations, and results.
>
>
>
> We also notice that the orca paper also uses GPT-4 to generate CoT data. We will cite this.
>
>
>
> As for the SuperCoT (https://huggingface.co/ausboss/llama-30b-supercot/tree/main) dataset, there isn't a dedicated paper available, nor could we locate any detailed technical reports discussing its intricacies. Thus we have added a link and a brief discussion.
>
>
>
> Overall, while the arena of instruction tuning LMs with CoT data is burgeoning with fresh ideas and methodologies, LogiCoT was, to our knowledge, the first to harness logical reasoning rationales through GPT-4 and logical reasoning datasets, leading to enhanced understanding and performance in logical reasoning tasks.
>
>
>
> **3. Clarity in the Experiments Section:**
>
> > 3. The experiment section is not very clearly described. The LogiEval benchmark could be explained further, for example, AL:AR-LSAT,CT:ConTRoL,HL:HELP,TN:TaxiNLI tasks are never introduced in the paper. The LogiEval benchmark used for evaluation contains two datasets identical from fine-tuning data: LogiQA, ReClor. However, the authors don't address this data contamination issue. The authors also don't mention how the evaluation tasks were inputted: were they also using an instruction format? Perhaps the performance gains stem from instruction understanding (which smaller models are very sensitive to) and not from reasoning improvements. Authors don't conduct any evaluation of the reasoning chains themselves, but rather only of the final answer accuracy. Without this information, I believe the central claim of the paper that "LogiCoT ... elicits general reasoning skills" (line 017-018) is not sufficiently substantiated.
>
>
>
> **Response:** We appreciate the feedback on the clarity of the experiments section.
>
>
>
> ​                ● *Description of LogiEval and Tasks:* We provide a detailed introduction and description of each task (AL: AR-LSAT, CT: ConTRoL, HL: HELP, TN: TaxiNLI) in the revised manuscript for clarity.
>
>
>
> ​                ● *Data Contamination Issue:* We understand the concerns regarding potential data contamination from using LogiQA and ReClor datasets in both training and evaluation. The potential for data contamination between training and evaluation is a pertinent concern. In our work, we have taken care to ensure that the data used for training and evaluation do not overlap, thereby minimizing any bias. Specifically, we utilized the training sets of LogiQA and ReClor for model fine-tuning, and evaluations were exclusively conducted on their respective test sets. This distinction ensures the evaluation's integrity and provides a genuine measure of the model's performance on unseen data.
>
>
>
> ​                ● *Input Format of Evaluation Tasks:* The tasks were indeed inputted using an instruction format, similar to the one used during training. We agree that performance can be influenced by instruction understanding, especially for smaller models. To delve deeper into this, we conduct ablation studies by evaluating the model's performance with and without instructions. This would help tease apart the contribution of instruction understanding from actual reasoning improvements.
>
> The table below shows the results of LLaMA-7b-LogiCoT on the LogiEval benchmark with and without instructions.
>
> | Instruction | LQ   | LQ zh | RC   | AL   | LQ ood | CT   | HL   | TN   |
> | ----------- | ---- | ----- | ---- | ---- | ------ | ---- | ---- | ---- |
> | with        | 51.2 | 33.4  | 57.9 | 20.9 | 38.9   | 35.4 | 35.8 | 43.3 |
> | without     | 49.1 | 32.8  | 57.2 | 20.6 | 38.8   | 30.7 | 35.4 | 38.5 |
>
> Overall, the performance of LLaMA-7b-LogiCoT is not affected significantly by the absence of instructions.
>
>
>
> ​                ● *Evaluation of Reasoning Chains:*
>
>
> To ascertain the competence of the LLaMA-7b-LogiCoT in generating reasoning chains, we employed a similar evaluation methodology as we did for our LogiCoT data. Using the LogiEval benchmark as a yardstick, we examined a subset of 160 instances, obtained by picking 20 output examples from each dataset.
>
> The ensuing evaluation revealed that LLaMA-7b-LogiCoT performed commendably, achieving a mean score of 4.7 in Relevance, 4.5 in Coherence, 4.0 in Completeness, and 4.4 in Faithfulness.
>
> The table below compares the output quality of LLaMA-7b-LogiCoT with other popular models.
> | Model   | Coherence | Completeness | Faithfulness | Relevance |
> | ------- | --------- | ------------ | ----------- | --------- |
> | LLaMA-30b-supercot   | 3.4       | 3.5          | 3.1         | 4.5       |
> | Falcon-40b-instruct  | 3.2       | 3.4          | 3.1         | 4.5       |
> | ChatGPT | 3.9       | 4.8          | 3.7         | 4.7       |
> | GPT-4   | 4.5       | 4.8          | 4.5         | 4.9       |
> |LLaMA-7b-LogiCoT | 4.5 | 4.0 | 4.4 | 4.7 |
>
> In juxtaposition with our LogiCoT data assessment, there are evident distinctions. While the LLaMA-7b-LogiCoT trails slightly in categories like Relevance, Coherence, and Faithfulness, a more pronounced gap of 0.5 is discernible in the Completeness category. The drop in the Completeness score indicates smaller models are not competitive with GPT-4 on the length of outputs. However, compared to other community models, the quality of the reasoning chain yield by LLaMA-7b-LogiCoT is competitive, given that it is a much smaller model.
>
>
>
> **Typos, Grammar, Style, and Presentation Improvements:**
>
>
>
> > ● On Line 489: "language learning models" -> "large language models?"
>
>   **Response:** The typo has been corrected in the manuscript.
>
>
>
> > ● Regarding the numerous figures in the abstract (do you mean appendix?) and referenced verbiage on lines 378-386:
>
>   **Response:** We have restructured the relevant section, ensuring a smoother flow of information.

---

### Official Review · Reviewer_D4Qn · 2023-08-12

**Soundness:** 3

**Excitement:**

4: Strong: This paper deepens the understanding of some phenomenon or lowers the barriers to an existing research direction.

**Paper Topic And Main Contributions:**

The paper presented an instruction-tuning dataset with 70K instances, by collecting chain-of-thought logical reasoning responses from GPT4 based on existing datasets. The instruction focused on logical inference, symbolic logic, and reading comprehension.

The model trained on this dataset had large performance improvement on the reasoning benchmarks compare with the baseline.

**Questions For The Authors:**

A. In the experiment, how did you prompt the instruction-tuned models and how is the answer extracted?

B. Was the LogiQA Chinese version included in the dataset? If not it’s interesting to see the large improvement on LQ zh, can you show some example reasoning chains on that dataset?

C. It would be good to provide hyperparameters used for training LLaMA-7b-logicot for better reproducibility.

**Reasons To Accept:**

- A good instruction-tuning dataset is something the community eagerly need in order to close the instruction-following ability gap between open source models and close models. On top of that, the paper considered the logical reasoning aspect, which is new and has the potential to enable more interesting model applications.
- The paper spent good effort on designing various instructions that includes various important aspects of logical reasoning.
- The performance of resulting model on MMLU is boosted significantly, which is a good sign that the dataset is helpful for learning general reasoning ability and can preserve past knowledge.

**Reasons To Reject:**

It’s concerning that while all other reading comprehension datasets get significant improvement, the improvement on AL isn’t clear (where ChatGPT and GPT-4 still show similar performance advantage). Since LQ and RC are included in the training set, it’s concerning that AL’s result is an outlier, despite it should benefits a lot from the type of reasoning included in the instruction set. Can the author give some explanation on why that happens? A better experiment to run is to remove RC from the training set and evaluate on LQ, and vice versa, to make sure the model trained on the dataset can generalize well.

**Reproducibility:**

4: Could mostly reproduce the results, but there may be some variation because of sample variance or minor variations in their interpretation of the protocol or method.

**Reviewer Confidence:**

3: Pretty sure, but there's a chance I missed something. Although I have a good feel for this area in general, I did not carefully check the paper's details, e.g., the math, experimental design, or novelty.

---

> ### Author Rebuttal · Authors · 2023-08-28
>
> Thank you for your insightful comments. We appreciate the opportunity to further clarify our methodologies and decisions. Below are the elaborations to your questions:
>
> ​            A.     **Prompting and Answer Extraction from Instruction-Tuned Models:**
>
> > A. In the experiment, how did you prompt the instruction-tuned models and how is the answer extracted?
>
> **Response:** We will add relevant details to the Appendix. Below we list the final chosen prompts for each of the tasks.
>
> **LogiEval** is an instruction-prompting style benchmark where each dataset instance has integrated instructions. This ensures the LLM comprehends the desired format and response requirements.
>
> Below is the LogiEval instruction for the reading comprehension task:
>
> ```Instructions: You will be presented with a passage and a question about that passage. There are four options to be chosen from, you need to choose the only correct option to answer that question. If the first option is right, you generate the answer 'A', if the second option is right, you generate the answer 'B', if the third option is right, you generate the answer 'C', if the fourth option is right, you generate the answer 'D', if the fifth option is right, you generate the answer 'E'. Read the question and options thoroughly and select the correct answer from the four answer labels. Read the passage thoroughly to ensure you know what the passage entails.```
>
> And the instructions for the NLI task:
>
> ```Instructions: You will be presented with a set of facts and rules as premises, and a hypothesis about it. You need to decide whether the hypothesis is entailed by the premise by choosing one of the following answers: 'Yes': The hypothesis follows logically from the information contained in the premise. 'No': The hypothesis is logically false from the information contained in the premise. 'Neutral': It is not possible to determine whether the hypothesis is true or false without further information. Read the passage of information thoroughly and select the correct answer from the three answer labels. Read the premise thoroughly to ensure you know what the premise entails.```
>
> In the case of the **MMLU dataset**, we remain consistent with the protocols from previous evaluations. The prompt, "The following are multiple choice questions (with answers) about {subject}. {question} Answer:", provides both context and a clear response expectation.
>
> Our evaluation approach emphasizes exact matching. We instruct LLMs to generate the precise answer in accordance with the given prompt. The first token of the output is deemed as the final answer, which tests the LLM's proficiency in following instructions.
>
> ​            B.     **Inclusion of LogiQA Chinese Version:**
>
> > B.     Was the LogiQA Chinese version included in the dataset? If not it’s interesting to see the large improvement on LQ zh, can you show some example reasoning chains on that dataset?
>
> **Response:** Indeed, we integrated 1683 instances from the training set of **LogiQA Chinese version** as our seeding data. Given LogiQA's origin as a logical reasoning dataset curated from the Chinese internet, it inherently exists in Chinese. Thus, the Chinese version of the dataset was our default choice.
>
> We've undertaken refinements to our paper to enhance clarity. Our script at L260 has been revised from "We use the training set of the two datasets, keeping the test set out of the instruction tuning data." to "We use the training set of the two datasets, excluding the test set from the instruction tuning data. The Chinese version of the LogiQA dataset is used."
>
> ​            C.     **Hyperparameters for Training LLaMA-7b-logicot:**
>
> > C.     It would be good to provide hyperparameters used for training LLaMA-7b-logicot for better reproducibility.
>
> **Response:** We acknowledge the significance of reproducibility. Following the excellent work of the Stanford Alpaca (https://github.com/tatsu-lab/stanford_alpaca), we adopted their running scripts and set the hyper-parameters as follows:
>
> ​                ● learning rate: 2e-5
>
> ​                ● batch size: 4
>
> ​                ● training epochs: 2
>
> We have added the above details in our revision. In line with this, we are fully prepared to open-source our training script along with the specific hyperparameters used.
>
> ##### **Response to "Reasons to Reject":**
> > **Reasons To Reject:**It’s concerning that while all other reading comprehension datasets get significant improvement, the improvement on AL isn’t clear (where ChatGPT and GPT-4 still show similar performance advantage). Since LQ and RC are included in the training set, it’s concerning that AL’s result is an outlier, despite it should benefits a lot from the type of reasoning included in the instruction set. Can the author give some explanation on why that happens? A better experiment to run is to remove RC from the training set and evaluate on LQ, and vice versa, to make sure the model trained on the dataset can generalize well.
>
> **Response:** We recognize the concerns about the improvement in the AL dataset, particularly when benchmarked against the other reading comprehension datasets. We have in our revision added several potential reasons that might explain this variance:
>
> ​                ● As you rightly pointed out, the LQ and RC datasets operate on a 4-option multiple-choice format, whereas the AL dataset has a 5-option format. The answer might be biased toward the first 4 choices, A, B, C, and D, which frequently appear in the training corpus. This structural difference can introduce added complexity for the model, as the potential solution space for each question is larger in AL, making the task inherently more challenging.
>
> ​                ● Even beyond the format difference, the intrinsic nature of AL questions could differ from LQ and RC. While they have been included in the training set, there might be specific nuances within the AL dataset that aren't well-covered in the instruction set, affecting performance.
>
> To understand more about the roles of different types of training data, we conducted an ablation study as follows:
>
> **Ablation Study**
>
> To better understand the specific contributions of various reasoning types employed in our instruction-tuning process, we conduct an ablation study. This involves evaluating model performance by selectively removing one reasoning type at a time and observing the resultant change in the model's ability to handle logical reasoning tasks.
>
> The ablation study uses the instruction-tuned LLaMA-7b model. For each reasoning type, we train the model without that specific type while retaining the others. This is performed iteratively for all reasoning types.
>
> We report the overall average score of LogiEval and MMLU, respectively, to provide a comprehensive understanding of the respective impacts of these reasoning types. The results are shown in Table.
>
> | Removed            | LogiEval | MMLU |
> | ------------------ | -------- | ---- |
> | None (Full data)   | 40.7     | 43.3 |
> | Language to Logic  | 32.4     | 38.5 |
> | One-step Inference | 38.1     | 37.7 |
> | Inference chain    | 30.8     | 35.0 |
> | Multi-choice       | 35.6     | 30.9 |
>
> Below is our detailed analysis:
>
> **Language to Logic:** Excluding this type led to a decline in performance on the LogiEval dataset. This underscores its significance in mapping linguistic constructs to logical expressions, which is foundational for reasoning tasks.
>
> **One-step Inference:** The drop in accuracy suggests that even simple inferences play a vital role, especially in tasks where direct conclusions are drawn from given premises.
>
> **Inference Chain:** The model's performance drop on both datasets highlights the importance of chained logical deductions. Tasks that require multi-step reasoning particularly benefit from this type.
>
> **Multi-choice:** Removing multi-choice reasoning impacted performance on MMLU more than on LogiEval, emphasizing its role in tasks where choosing among alternatives based on logical grounds is essential.
>
>
> This ablation study reaffirms the unique contributions of each reasoning type to the model's performance. While all reasoning types contribute to enhancing the model's logical understanding, their impacts vary based on the nature of tasks and the datasets used. Future research can delve deeper into optimizing instruction-tuning processes based on specific reasoning type requirements of datasets.
>
> Apart from the ablation study, we also added data quality human evaluation to our LogiCoT data.
> we conducted a comprehensive human evaluation of the generated reasoning chains to assess their quality, faithfulness, and diversity.
>
> The evaluation procedure can be described as follows:
>
> 1: **Sampling Data:** Randomly sampled a subset of the generated reasoning chains for evaluation. We made sure this sample is representative of the entire dataset by randomly selecting 50 instances from each of our defined instruction types.
>
> 2: **Setting Evaluation Metrics:** For reasoning chains, we have considered:
>
>  ● **relevance:** Does the chain directly relate to the question?
>
>  ● **coherence:** Is the chain logically consistent?
>
>  ● **completeness:** Does it offer a full explanation for the reasoning?
>
>  ● **faithfulness:** Is the reasoning factual and not fabricating details?
>
> 3: **Evaluation Protocol:** We provided annotators with clear guidelines regarding each metric. They should rate each reasoning chain according to the metrics on a scale of 1 to 5.
>
> 4: **Analysis:** We calculated the average scores and inter-annotator agreement using Cohen's Kappa to ensure consistency among annotators.
>
> We outsourced 3 annotators to rate the quality and relevance of the generated chains. They are assigned with the same evaluation set, each sample is assigned with 3 scores to ensure reliability. The process took two days.
>
> The evaluations, coupled with their findings, are added to the revised version of the paper, fortifying our claims.
>
> The added human evaluation results are as follows:
>
> **Human Evaluation of Dataset Quality**
>
> We conducted a comprehensive human evaluation of our generated reasoning chains. A random subset of 200 reasoning chains was selected. These chains were evaluated by 3 domain professional annotators using four key metrics: Relevance (Does the chain directly relate to the question?), Coherence (Is the chain logically consistent?), Completeness (Does it offer a full explanation for the reasoning?), and Faithfulness ( Is the reasoning factual and not fabricating details?). Each reasoning chain was rated on a scale from 1 (poor) to 5 (excellent) for each metric.
>
> The reasoning chains achieved **an average score of 4.9 for Relevance, 4.7 for Coherence, 4.5 for Completeness, and 4.5 for Faithfulness**. The inter-annotator agreement, measured using Cohen's Kappa, was 0.87, indicating strong agreement among the annotators.
>
> The human evaluation underscores the quality and reliability of the reasoning chains generated by our model. By conducting new experiments and evaluations to our method, we'd like to increase the soundness of our paper.

---

### Meta-Review · Area_Chair_ZDDG · 2023-09-20

**Recommendation:** 3

**Metareview:**

This paper investigated in the construction of a new instruction tuning dataset for logical chain-of-thought reasoning with GPT-4. The authors showed the competitive logical reasoning and general inference abilities with the proposed approach. This paper addressed an important research problem for instruction tuning of LLM with logical reasoning. The reviewers found the experiment results strong and convincing. On the down side, two reviewers expressed concerns on the experiment setting, related to choice of reasoning types, comparison to baseline models, and the potential data contamination. One reviewer commented on the lack of the evaluation on the quality of the proposed dataset. The authors were able to provide the human evaluation of the dataset which the author should consider add to the revised version of the paper. The paper presentation also has room for improvement as commented by two reviewers.

---

### Decision · Program_Chairs · 2023-10-07

**Decision:**

Accept-Findings

**Comment:**

This paper investigated in the construction of a new instruction tuning dataset for logical chain-of-thought reasoning with GPT-4. The authors showed the competitive logical reasoning and general inference abilities with the proposed approach. This paper addressed an important research problem for instruction tuning of LLM with logical reasoning. The reviewers found the experiment results strong and convincing. On the down side, two reviewers expressed concerns on the experiment setting, related to choice of reasoning types, comparison to baseline models, and the potential data contamination. One reviewer commented on the lack of the evaluation on the quality of the proposed dataset. The authors were able to provide the human evaluation of the dataset which the author should consider add to the revised version of the paper. The paper presentation also has room for improvement as commented by two reviewers.